



# Foraminiferal community response to seasonal anoxia in Lake Grevelingen (the Netherlands)

Julien Richirt[1], Bettina Riedel[1,2], Aurélia Mouret[1], Magali Schweizer[1], Dewi Langlet[1,3], Dorina Seitaj[4], Filip J. R. Meysman[5,6], Caroline P. Slomp[7] and Frans J. Jorissen[1]

[1]UMR 6112 LPG-BIAF Recent and Fossil Bio-Indicators, University of Angers, 2 Boulevard Lavoisier, F-49045 Angers, France
[2]First Zoological Department, Vienna Museum of Natural History, Burgring 7, 1010 Vienna, Austria
[3]UMR 8187, LOG, Laboratoire d'Océanologie et de Géosciences, University of Lille, CNRS, University of Littoral Côte d'Opale, F 62930 Wimereux, France
[4]Department of Ecosystem Studies, Royal Netherlands Institute for Sea Research (NIOZ), Yerseke, the Netherlands
[5]Department of Biology, University of Antwerp, Universiteitsplein 1, BE-2610 Wilrijk, Belgium
[6]Department of Biotechnology, Delft University of Technology, 2629 HZ Delft, the Netherlands
[7]Department of Earth Sciences (Geochemistry), Faculty of Geosciences, Utrecht University, Princetonlaan 8a, 3584 CB Utrecht, the Netherlands

*Correspondence to*: Julien Richirt (richirt.julien@gmail.com)

**Abstract.** Over the last decades, hypoxia in marine coastal environments have become more and more widespread, prolonged and intense. These hypoxic events have large consequences for the functioning of benthic ecosystems. They profoundly modify early diagenetic processes involved in organic matter recycling, and in severe cases, they may lead to complete anoxia and presence of toxic sulphides in the sediment and bottom water, thereby severely affecting biological compartments of benthic marine ecosystems. Within these ecosystems, benthic foraminifera show a high diversity of ecological responses, with a wide range of adaptive life strategies. Some species are particularly resistant to hypoxia/anoxia and consequently, it is interesting to study the whole foraminiferal community as well as species specific responses to such events. Here we investigated the temporal dynamics of living benthic foraminiferal communities (recognised by CellTracker™ Green) at two sites in the saltwater Lake Grevelingen in the Netherlands. These sites are subject to seasonal anoxia with different durations and are characterised by the presence of free sulphide ($H_2S$) in the uppermost part of the sediment. Our results indicate that foraminiferal communities are impacted by the presence of $H_2S$ in their habitat, with a stronger response in case of longer exposure times. At the deepest site (34 m), one to two months of anoxia and free $H_2S$ in the surface sediment resulted in an almost complete disappearance of the foraminiferal community. Conversely, at the shallower site (23 m), where the duration of anoxia and free $H_2S$ was shorter (one month or less), a dense foraminiferal community was found throughout the year. Interestingly, at both sites, the foraminiferal community showed a delayed response to the onset of anoxia and free $H_2S$, suggesting that the combination of anoxia and free H2S does not lead to increased mortality, but rather to strongly decreased reproduction rates. At the deepest site, where highly stressful conditions prevailed for one to two months, the recovery time of the community takes about half a year. In Lake Grevelingen, *Elphidium selseyense* and *Elphidium magellanicum* are much





less affected by anoxia and free H₂S than *Ammonia* sp. T6. We hypothesise that this is not due to a higher tolerance of $H_2S$, but rather related to the seasonal availability of food sources, which could have been less suitable for *Ammonia* sp. T6 than for the elphidiids.

## 1 Introduction

Hypoxia affects numerous marine environments, from the open ocean to coastal areas. Over the last decades, a general decline

in oxygen concentration was observed in marine waters (Stramma et al., 2012), with an extent varying between the concerned regions. In coastal areas, oxygen concentrations have been estimated to decrease 10 times faster than in the open ocean, with indications of a recent acceleration, expressed by increasing frequency, intensity, extent and duration of hypoxic events (Diaz and Rosenberg, 2008; Gilbert et al., 2010). The combination of global warming and eutrophication is strengthening seasonal stratification of the water column, decreasing oxygen solubility, and enhancing benthic oxygen consumption in response to

increased primary production, resulting from increased anthropogenic nutrient and/or organic matter input (i.e. eutrophication, Diaz and Rosenberg, 2008). Bottom water hypoxia has serious consequences for the functioning of all benthic ecosystem compartments (see Riedel et al., 2016 for a review). Benthic faunas are strongly impacted by these events (Diaz and Rosenberg, 1995) although the meiofauna, especially foraminifera, appears to be less sensitive to low Dissolved Oxygen (DO) concentrations than the macrofauna (e.g. Josefson and Widbom, 1988). Many foraminiferal taxa are able to withstand seasonal

hypoxia/anoxia (e.g.  Alve and Bernhard, 1995; Moodley et al., 1997, 1998a; Geslin et al., 2004; Pucci et al., 2009; Koho et al., 2011; Langlet et al., 2013), and consequently can play a major role in carbon cycling in ecosystems affected by seasonal low oxygen concentrations (Woulds et al., 2007). Anoxia is often accompanied by free sulphide (H₂S) in pore and/or bottom waters, which is considered very harmful for the benthic macrofauna (Wang and Chapman, 1999). Neutral molecular H₂S can diffuse through cellular membranes and inhibits the functioning of cytochrome *c* oxydase (a mitochondrial enzyme involved

in ATP production), finally inhibiting aerobic respiration (Nicholls and Kim, 1982; Khan et al., 1990; Dorman et al., 2002). Lake Grevelingen (southwestern Netherlands) is a former branch of the Rhine-Meuse-Scheldt estuary, which was closed in its eastern part (riverside) by the Grevelingen Dam in 1964 and in its western part (seaside) by the Brouwers Dam in 1971. The resulting saltwater lake, with a surface of 115 km², is one of the largest saline lakes in Western Europe. Lake Grevelingen is characterised by a strongly reduced circulation (even after the construction of a small sluice in 1978) with a strong thermal

stratification occurring in the main channels in summer, leading  to seasonal bottom water hypoxia/anoxia in late summer and early autumn (Bannink et al., 1984). This situation leads to a rise of the H₂S front in the uppermost part of the sediment, sometimes up to the water-sediment interface.

These observations especially concern the Den Osse Basin (i.e. one of the deeper basins, maximum depth 34 m; Hagens et al., 2015), which has been intensively monitored over the last decades, so that a large amount of environmental data is available

(e.g. Wetsteijn, 2011; Donders et al., 2012). The annual net primary production in the Den Osse Basin (i.e. 225 g C m⁻² y⁻¹, Hagens et al., 2015) is comparable to other estuarine systems in Europe (Cloern et al., 2014). However, there is almost no





nutrient input from external sources, thus primary production is largely based on autochthonous recycling (>90 %, Hagens et al., 2015), both in the water column and in the sediment, with a very strong pelagic/benthic coupling (de Vries and Hopstaken, 1984). The benthic environment is characterised by the presence of two antagonistic groups of bacteria, with contrasting
seasonal population dynamics (i.e. cable bacteria in winter/spring and *Beggiatoaceae* in autumn/winter), which have a profound impact on all biogeochemical cycles in the sediment column (Seitaj et al., 2015; Sulu-Gambari et al., 2016a, 2016b). The combination of hypoxia/anoxia with sulphidic conditions, which is rather unusual in coastal systems without external nutrient input, and the activity of antagonistic bacterial communities make Lake Grevelingen (and especially the Den Osse Basin) a very peculiar environment. In the Den Osse Basin, seasonal anoxia coupled with $H_2S$ presence at or very close to the
water-sediment interface occur in summer. However, euxinia (i.e. diffusion of free $H_2S$ in the water column) does not occur, because of the cable bacterial activity (Seitaj et al., 2015).

Although the large tolerance of foraminifera to low DO contents is well known (e.g. Moodley and Hess, 1992; Alve and Bernhard, 1995; Moodley et al., 1997; Geslin et al., 2004; Pucci et al., 2009; Koho et al., 2011; Langlet et al., 2013), their tolerance to free $H_2S$ is still debated  (Bernhard, 1993; Moodley et al., 1998b; Panieri and Sen Gupta, 2008; Langlet et al.,
2014). Moreover, studies on foraminiferal population dynamics in systems affected by seasonal hypoxia/anoxia with sulphidic conditions are still very sparse. The highly peculiar environmental context of Lake Grevelingen offers an excellent opportunity to study this still poorly known aspect of foraminiferal ecology. Previous studies of living benthic foraminifera in environments subjected to hypoxia/anoxia were almost all based on Rose Bengal stained samples (e.g. Gustafsson and Nordberg, 1999, 2000; Duijnstee et al., 2004; Panieri, 2006; Schönfeld and Numberger, 2007; Polovodova et al., 2009; Papaspyrou et al., 2013).
However, foraminiferal protoplasm may remain stainable from several weeks to months after their death (Corliss and Emerson, 1990), especially under low dissolved oxygen concentrations where organic matter degradation may be very slow (Bernhard, 1988; Hannah and Rogerson, 1997; Bernhard et al., 2006). The Rose Bengal staining method is therefore not suitable for studies in environments affected by hypoxia/anoxia. Consequently, the results of foraminiferal studies in low oxygen environments based on this method have to be considered with reserve. In order to avoid this problem, we used CellTracker™
Green (CTG) to recognise living foraminifera. CTG is a fluorescent probe which marks only living individuals with cytoplasmic (i.e. enzymatic) metabolic activity (Bernhard et al., 2006). Since metabolic activity stops after the death of the organism, CTG should give a much more accurate assessment of the living assemblages at the various sampling times.

In this study, samples were collected in August and November 2011 and then every month through the year 2012, at two different stations in the Den Osse Basin, with two replicates dedicated to foraminifera. The two stations were chosen in
contrasted environments regarding water depth (34 m and 23 m, respectively) and duration of seasonal hypoxia/anoxia and sulphidic conditions. Foraminiferal assemblages were studied in the top 1 cm layer. For each dominant species, size distributions were determined in order to get insight into the population dynamics. The seasonal variability study of the foraminiferal community allows us (1) to better understand the foraminiferal tolerance to seasonal hypoxia/anoxia with presence of free $H_2S$ in their microhabitat and (2) to obtain information about the life histories of the various species under
adverse conditions. This knowledge will be useful for the development of indices assessing environmental quality (i.e.



biomonitoring) and may also improve paleoecological interpretations of coastal records (e.g. Murray, 1967; Gustafsson and Nordberg, 1999).

## 2 Material and Methods

Lake Grevelingen is a part of the former Rhine-Meuse-Scheldt estuarine, in the southwestern Netherlands. This former
estuarine branch was turned into an artificial saltwater lake during the Delta Works project. Due to the thermal stratification and high oxygen consumption in the benthic compartment, development of bottom water hypoxia/anoxia occurs in the deepest part of the basin in summer to early autumn (Bannink et al., 1984; Hagens et al., 2015). In the literature, the terminology and threshold values used to describe oxygen depletion are highly variable (e.g., oxic, dysoxic, hypoxic, suboxic, microxic, postoxic; see Jorissen et al., 2007; Altenbach et al., 2012). In this study we defined hypoxia by a concentration of oxygen <63
µmol L$^{-1}$ (1.4 mL L$^{-1}$ or 2 mg L$^{-1}$) whereas anoxia is defined as no detected oxygen (following Rabalais et al., 2010).

### 2.1 Environmental parameters

The two studied sites are located along a depth gradient in the Den Osse Basin of Lake Grevelingen. Both station 1 (51°44.834' N, 3°53.401' E) and station 2 (51°44.956' N, 3°53.826' E) are located in the main channel, at 34 and 23 m depth, respectively (see Petersen et al., 2019).
Measurements of oxygen concentrations in the bottom water (1 m above the sediment-water interface using a CTD) for 2011 are from Donders et al. (2012), whereas the 2012 data are from Hagens et al. (2015) and Seitaj et al (2017). Oxygen Penetration Depth (OPD) and depth of free H$_2$S detection were determined using O$_2$ and H$_2$S microsensors by Seitaj et al., (2015) for station 1, and the data for station 2 were acquired similarly (Supplementary Table 1).

### 2.2 Field Sampling

Two replicate sediment cores (inner diameter 6 cm) dedicated to the foraminiferal study were sampled in August and November 2011 and then monthly throughout the year 2012. The uppermost centimetre of each core was labelled with CellTracker™ Green CMFDA (CTG, 5-chloromethylfluorescein diacetate, final concentration of 1µM following Bernhard et al., 2006) and fixed in 5 % sodium borate buffered formalin after 24 h of incubation. Since picking foraminifera under an epifluorescence stereomicroscope is particularly time-consuming, we decided to study samples only every two months for the
year 2012. At a later stage, in view of the large differences in foraminifera abundances between the samples of September and November 2012 at station 2, we decided to study the October and December 2012 samples as well for this station. The sampling dates investigated in this study are listed in Table 1.



### 2.3 Sample Treatment

All samples were sieved over 315, 150, 125 µm meshes, and foraminiferal assemblages were studied in all three size fractions.
Individuals were picked wet under an epifluorescence stereomicroscope (Olympus SZX12, light fluorescent source Olympus
URFL-T, excitation/emission wavelengths: 492 nm/517 nm) and placed on micropalaeontological slides. Only specimens that
fluoresced brightly green were considered as living and were identified to the (morpho-)species level when possible.
Abundances were then standardised to a volume of 10 cm$^3$ in order to facilitate comparison with previous studies. The
abundances of living foraminifera for each sampling time and replicate are listed in Supplementary Tables 2 and 3. The mean
abundance and standard deviation for the two replicates for each sampling date were calculated both for the total living
assemblage and the individual species, as an indication of spatial patchiness.

### 2.4 Taxonomy of dominant species

Four dominant species (>1 % of the total assemblage) were present in our material: *Ammonia* sp. (T6), *Elphidium
magellanicum* (Heron-Allen and Earland, 1932), *Elphidium selseyense* (Heron-Allen and Earland, 1911) and *Trochammina
inflata* (Montagu, 1808). As we identified these species on the basis of morphological criteria, we will use them as
"morphospecies".

Concerning the genus *Ammonia*, two living specimens collected at Grevelingen  station 1 were molecularly identified (by
DNA barcoding) as phylotype T6 by Bird et al. (2019). At the same site, we genotyped seven other living *Ammonia* specimens,
which were all T6. Their sequences were deposited on GenBank (accession numbers MN190684 to MN190690) and
Supplementary Figure 1 shows Scanning Electron Microscope (SEM) images of the spiral side and of the penultimate chamber
at 1000x magnification for four individuals. A morphological screening based on the criteria proposed by Richirt et al. (2019)
confirmed that T6 accounts for the vast majority (>98 %) of *Ammonia* individuals, whereas T1, T2, T3 and *Ammonia
falsobeccarii* are only present in very small amounts (Supplementary Table 3).

The specimens of *Elphidium magellanicum* were identified exclusively on the basis of morphological criteria, as there are no
molecular data available yet. This morphospecies, although rare, is regularly recognised in boreal and lusitanean provinces of
Europe (e.g. Gustafsson and Nordberg, 1999; Darling et al., 2016; Alve et al., 2016). However, as the type species was
described from the Magellan strait (Southern Chile), the European specimens may represent a different species and further
studies involving DNA sequencing of both populations are needed to confirm or infirm this taxonomic attribution (see Roberts
et al., 2016).

In the past, *Elphidium selseyense* has often been considered as an ecophenotype of *Elphidium excavatum* (Terquem, 1875) and
has been identified as *E. excavatum* forma *selseyensis* (e.g. Feyling-Hanssen, 1972; Miller et al., 1982). Recently, Darling et
al. (2016) showed that the various ecophenotypes recognised in *E. excavatum* are in fact genetically separated and therefore
represent different species. Four living specimens of the *E. excavatum* group sampled at station 1 for DNA analysis were all
identified as *E. selseyense* (phylotype S5, Darling et al., 2016). We only observed minor morphological variations in our



material, especially concerning the number of small bosses in the umbilical region, which we considered as intraspecific variability. Consequently, we identified all our specimens as *E. selseyense*.

The specimens attributed to *Trochammina inflata* were also identified exclusively on the basis of morphological criteria, as no molecular data are available yet.

## 2.5 Size distribution measurement

In order to gain insight into the foraminiferal population dynamics, size measurements were performed on all samples of 2012. Prior to measurements, trochospiral species were all orientated in the same way (spiral side up). High-resolution images (3648*2736 pixels) of all micropalaeontological slides were taken with a stereomicroscope (Leica S9i, 10x magnification). In order to obtain measurements for all individual specimens, images were processed using ImageJ software (Schneider et al., 2012, Fig. 1).

The three size fractions (125–150, 150–315, >315 µm) were analysed together for the size distribution analyses. For each individual, the maximum diameter was measured (i.e. Feret's diameter). We represented all size distributions using histograms with 20 µm classes (the best compromise between the total number of individuals and the size range). In order to compare more easily months and species, the median and the mode (associated with the numbers of individuals) were calculated for each size distribution. As we only examined the size fractions >125 µm, our analysis mainly concerns adult specimens, and

does not include juveniles. This limitation should be kept in mind when interpreting the results.

## 3 Results

### 3.1 Total abundances of foraminiferal assemblages

Figure 2 shows the total living foraminiferal abundance for each replicate, and the mean and standard deviation computed for the two replicates ($\overline{x} \pm sd$) of the 0–1 cm depth interval for the two studied stations. Total abundances varied between $1.1 \pm$

$1.5$ and $449.9 \pm 322.1$ ind. 10 cm$^{-3}$ for station 1, and between $91.1 \pm 25$ and $604.8 \pm 3.5$ ind. 10 cm$^{-3}$ for station 2. For every studied month, the total density was higher at station 2 than at station 1. The seasonal succession is very different between the two sites. Station 1 shows very low total foraminiferal abundances for most months, contrasting with a much higher density in late spring (May) and early summer (July). Conversely, station 2 shows high total foraminiferal abundances throughout the year, with somewhat lower values in late autumn (i.e. November 2011, October and November 2012).

At station 1, almost no individuals were present in August ($\overline{x} = 3.4 \pm 1.3$) and November 2011 ($\overline{x} = 1.1 \pm 1.5$). In 2012, total abundances were very low in January ($\overline{x} = 11.5 \pm 9.3$), showed a slight increase in March ($\overline{x} = 62.1 \pm 19.3$) and reached a maximal abundance in May ($\overline{x} = 449.9 \pm 322.1$). Total abundances then progressively decreased from May to September ($\overline{x} = 34 \pm 17$) and almost no foraminifera were present in November ($\overline{x} = 1.6 \pm 0.3$).





At station 2, total abundances were comparatively low in August and November 2011 ($\overline{x} = 174 \pm 48$ and $\overline{x} = 128.7 \pm 25$

ind. 10 cm$^{-3}$, respectively). In 2012, total abundances were relatively high and stable from January to September (between $\overline{x} = 523.6 \pm 30.7$ to $\overline{x} = 604.8 \pm 3.5$), then decreased in October ($\overline{x} = 211.5 \pm 8$) and November ($\overline{x} = 91.1 \pm 25.3$) and finally increased again in December ($\overline{x} = 377.9 \pm 38.8$).

### 3.2 Dominant Species

In this section, we will only consider the dominant morphospecies, which individually represent at least 1 % of the total

assemblage for the total assemblage sampled for each station (all samples taken together, Table 2).

At station 1, the major species were, in order of decreasing abundances, *Elphidium selseyense* (Fig. 3a–b), *Elphidium magellanicum* (Fig. 3c–d) and *Ammonia* sp. T6 (Fig. 3e–g). In Fig. 4, we added *Trochammina inflata* (Fig. 3h–j) to facilitate comparison with station 2, where this species is among the dominant ones. The "Other species" account only for 2.2 % of the total assemblage at station 1. The fact that they are well represented in some months (e.g. 26.3 % of the assemblage in August

2011) is due to the extremely low number of individuals (see Fig. 2 and Table 2). At station 2, the dominant species, in order of decreasing abundances, were *E. selseyense*, *Ammonia* sp. T6, *E. magellanicum* and *T. inflata* (Table 2). Here, "Other species" account only for 2.6 % of the total assemblage.

Whereas *E. selseyense* and *E. magellanicum* were dominant species at both stations, both *Ammonia* sp. T6 and *T. inflata* were present in much higher abundances at station 2 compared to station 1, where the latter species was almost absent (Fig. 4–5).

At station 1, only some very scarce individuals of *E. selseyense* and *Ammonia* sp. T6 were observed in August and November 2011 (Fig. 4). In 2012, *E. selseyense* and *E. magellanicum* together account always for 60 % or more of the fauna, except in January. The abundances of these two species were very low in January, started to increase in March ($\overline{x} = 23.9 \pm 6.8$ and $\overline{x} = 21.6 \pm 11$) to reach maximal values in May ($\overline{x} = 336.5 \pm 275.8$ and $\overline{x} = 96.4 \pm 47.3$). In July, values for *E. selseyense* were still high ($\overline{x} = 162 \pm 121.5$), whereas *E. magellanicum* had strongly decreased ($\overline{x} = 3.7 \pm 0.3$). Both species further

decreased until an almost total absence in November 2012. *Ammonia* sp. T6, started to be present with low abundances in January ($\overline{x} = 3.2 \pm 3.5$), to reach (fairly low) maximum abundances between March and July 2012 (ranging between $\overline{x} = 9.2 \pm 6.5$ and $\overline{x} = 12.9 \pm 1.3$). Then abundances rapidly decreased until the species was almost absent in November. *Trochammina inflata* was absent in 2011 and was only present with very low abundances from January to May and in November 2012.

At station 2, the two dominant major species were *E. selseyense* and *Ammonia* sp. T6 which together always represented at least 70 % of the total assemblage (Fig. 5 and Table 2). These two species showed a different seasonal pattern over the considered period. Abundances of *E. selseyense* were comparable in August ($\overline{x} = 74.8 \pm 29.8$) and November 2011 ($\overline{x} = 52.3 \pm 27$) then showed a progressive increase until a maximum in September 2012 ($\overline{x} = 365.5 \pm 70.3$). Abundances then showed a sharp decrease in October and November (respectively $\overline{x} = 98.7 \pm 8.5$ and $\overline{x} = 30.9 \pm 2.3$) to increase again in

December ($\overline{x} = 252.2 \pm 41$). For *Ammonia* sp. T6, abundances strongly increased between November 2011 ($\overline{x} = 60.8 \pm 1.5$)





and January 2012 ($\overline{x} = 226.2 \pm 52.3$) and then progressively decreased until the end of 2012 ($\overline{x} = 48.1 \pm 26$ in November 2012). *Trochammina inflata* showed a similar pattern as *Ammonia* sp. T6. Abundances strongly increased between November 2011 ($\overline{x} = 11.8 \pm 1.8$) and January 2012 ($\overline{x} = 121.5 \pm 29.8$), and then progressively decreased until very low abundances were found in November ($\overline{x} = 3.7 \pm 3$). *E. magellanicum* was completely absent in August and November 2011, almost absent

in January 2012 ($\overline{x} = 0.9 \pm 0.3$) and then suddenly increased until a maximum of $\overline{x} = 116 \pm 6.5$ in May. Conversely to station 1, abundances stayed relatively high in July ($\overline{x} = 37.8 \pm 2.5$) and September ($\overline{x} = 72 \pm 35.8$), and then drastically decreased until minimum numbers in October and November. Finally, like all other species, *E. magellanicum* abundances increased again in December ($\overline{x} = 25.5 \pm 13$).

### 3.3 Size distribution

In order to base our analysis on a sufficiently high number of specimens, we will here focus on *E. selseyense* and *Ammonia* sp. T6. As explained before, we will consider only specimens retained on a 125 µm mesh, which means that juvenile specimens are not represented. Only the samples taken in 2012 were considered.

The size distribution of *E. selseyense* was relatively similar between the two stations regarding the median, ranging from 253 µm (in May) to 295 µm (in November) at station 1 and from 261 µm (in October) to 290 µm (in March) at station 2. At both

stations, we observed the presence of an abundant group of smaller specimens, with a mode that never exceeded 250 µm, except in March at station 2, when it is difficult to separate this subpopulation from the larger specimens (Fig. 6). The main difference between the two stations is the higher proportion of larger individuals (>400 µm) at station 2, which is visible through the better-developed tails at the right side of the distribution graphs (Fig. 6).

The low number of *Ammonia* sp. T6 individuals at station 1 does not allow us to draw any firm conclusion concerning the size

distribution at this station. At station 2, a group of individuals with smaller diameters (< 300 µm) was always present (Fig. 7). The overall size distribution showed a clear shift to higher diameters between March (median = 279 µm) and May (median = 373 µm, Fig. 7), which is also evidenced by the much higher proportion of larger individuals. Specimens larger than 400 µm were abundantly found until November (median = 378 µm), but started to diminish in December, as is also shown by the decrease of the median to 339 µm.

Our tentative to distinguish cohorts by using a deconvolution method to separate the total size distributions into a sum of Gaussian curves was not conclusive. The main problem was the fact that we did not have any information concerning individuals smaller than 125 µm, so that our size distributions were systematically skewed on the left side (i.e. toward small individuals). An additional problem was the large number of smaller specimens which were always present. Because the identification of individual cohorts was not successful, parameters like reproduction rate, growth rate or lifespan were not

assessable. Nevertheless, the size distribution data give some clues concerning the population dynamics of the two dominant species.



### 3.4 Encrusted forms of *Elphidium magellanicum*

In our samples, during May at station 1 and May, July, September and December at station 2, we found abundant encrusted forms of *E. magellanicum* (Fig. 8). Most individuals were totally encrusted (Fig. 8a), others only partly (Fig. 8b). These crusts were hard, firmly stuck to the shell (difficult to remove with a brush), thin (Fig. 8c–e) and rather coarse (the crust seemed composed of sediment particles cemented by a rather homogenous matrix).

Because the crust stayed cohesive after exposition to 0.1 M of EDTA (EthyleneDiamineTetraacetic Acid) diluted in 0.1 M cacodylate buffer (acting as a carbonate chelator), it appears that this crust is composed of sediment particles cemented by an organic matrix. In view of the fact that the crusts consist mainly of organic matter, the encrusted individuals probably are specimens with preserved feeding cysts. Similar observations have been made for *Elphidium incertum* (Linke and Lutze, 1993; Gustafsson and Nordberg, 1999) and also in Flensburg Fjord, where partial cysts remained attached to the tests of *E. incertum* (Polovodova et al., 2009), similar to our observations (Fig. 8a).

Figure 9 shows the quantitative occurrence of encrusted specimens for the successive samples. At station 1, encrusted forms of *E. magellanicum* were present in moderate proportions in May (26.8 % of the total *E. magellanicum* population) and July (47.6 %); the species disappeared thereafter. At station 2, encrusted forms strongly dominated the *E. magellanicum* population from May (72.3 %) to December (88 %).

### 4 Discussion

### 4.1 Use of CellTracker™ Green

The conventional method to discriminate between live and dead foraminifera uses Rose Bengal, a compound which stains proteins (i.e. organic matter). This method was proposed for foraminifera by Walton (1952) and is based on the assumption that "*the presence of protoplasm is positive indication of a living or very recently dead organism*". The author already noted that this assumption implied that the rate of degradation of organic material should be relatively rapid. However, it appears that protoplasm degradation may be relatively long (from weeks to years, Corliss and Emerson, 1990), especially in hypoxic or anoxic conditions deeper in the sediment (Bernhard, 1988; Hannah and Rogerson, 1997). In these conditions, it can therefore not be excluded that dead individuals become stained as well. Bernhard et al. (2006) showed that abundances of living individuals recognised on the basis of Rose Bengal staining could be overestimated by a factor of two. The use of more trustworthy criteria is even more crucial in environments where organic matter may degrade very slowly, such as under low oxygen conditions. In this study, we used CellTracker™ Green (CTG), a fluorogenic probe (i.e. the substance becomes fluorescent after modification of the original molecule) which labels the enzymatic (esterase) activity in the foraminiferal cytoplasm (Bernhard et al., 2006). CTG allowed us to discriminate efficiently between living and dead foraminifera at the time of sampling, and to avoid over-estimation of the live foraminifera abundances.



## 4.2 Environmental setting of Den Osse Basin

At Lake Grevelingen, the water circulation was strongly limited by the construction of dams (in the early 1970s) and only a small sluice allows water exchanges with oceanic waters (i.e. very weak hydrodynamics). Nevertheless, in 2012, the

salinity ranged from 30 to 33. Consequently, Lake Grevelingen is euhaline and salinity variations are not likely to affect foraminiferal communities, since the dominant species (i.e. *E. selseyense, E. magellanicum* and *Ammonia* sp. T6) are known to be euryhaline (i.e. highly tolerant to salinity variations) and typically live in this salinity range (e.g. Bradshaw, 1957; Gustafsson and Nordberg, 2000; Murray and Alve, 2000; Darling et al., 2016; Mojtahid et al., 2016).

In Den Osse Basin, the nutrient input from external sources is very low and pelagic/benthic coupling is essential, as already

noted by de Vries and Hopstaken (1984). In 2012, phytoplankton blooms occurred in April-May and July (Hagens et al., 2015, Fig. 10) in response to the increasing solar radiation and the nutrient availability in the water column following organic matter recycling in winter. This led to an increased food availability in the benthic compartment in the same periods. In general, Chl *a* concentrations in Den Osse Basin are below 10 µg L$^{-1}$, excluding very short peaks during blooms in late spring (April–May) and summer (July) which didn't exceed 30 µg L$^{-1}$ in 2012 (Hagens et al., 2015). Thermal stratification of the water column

and increased oxygen consumption due to organic matter input (i.e. from phytoplankton blooms) are together responsible for the development of seasonal bottom water hypoxia/anoxia in summer. Although euxinia (i.e. diffusion of free $H_2S$ into the water column) does not occur in the Den Osse Basin due to cable bacterial activity in winter, free $H_2S$ is present in the uppermost layer of the sediment in summer (Seitaj et al., 2015). Summarising, in the benthic ecosystem, increased food availability in summer is counterbalanced by strongly decreasing oxygen contents, sometimes accompanied by the presence

of free sulphides in the topmost sediment. The tolerance of individual species to these conditions will influence their competitive success, which will ultimately control the community characteristics.

## 4.3 Foraminiferal tolerance to anoxia and free sulphide

Tolerance to long term anoxia (i.e. from weeks to 10 months) has been shown for many species of foraminifera from different types of environments (e.g. Bernhard, 1993; Bernhard and Alve, 1996; Moodley et al., 1997; Duijnstee et al., 2003, 2005;

Ernst et al., 2005; Pucci et al., 2009; Piña-Ochoa et al., 2010b; Langlet et al., 2013; Geslin et al., 2014). In the vast majority of these studies, no decrease in the total abundances of living foraminifera (i.e. strongly increased mortality) was observed during anoxic events. Unfortunately, observations concerning the foraminiferal tolerance to the presence of $H_2S$ in the sediment are much scarcer. The few available observations are not conclusive, but suggest that $H_2S$ could be toxic for foraminifera even on fairly short time scales.

Bernhard (1993) exposed diverse faunas collected at 23 m depth in Explorer's cove in Antarctica to euxinic conditions by using sealed flasks with seawater flushed with nitrogen and with a $H_2S$ concentration of 500 µmol L$^{-1}$. The author found that foraminiferal activity (as determined by ATP content) was not significantly affected after 30 days (32.6 ± 8.6 % of 174 ind. in control conditions and 29.5 ± 6.2 % of 173 ind. in sulphidic conditions). Conversely, for complete faunas from a 19 m deep





site in the Adriatic Sea, Moodley et al. (1998a) found a strong decrease of Rose Bengal stained foraminifera over the course

of the 66 days incubation in euxinic conditions (a maximum of $11.9 \pm 0.4$ µmol $L^{-1}$ of $H_2S$ in the overlying water). After 21 days, living specimens were still observed, whereas after 42 and 66 days, the live checks (based on protoplasm movement) gave only negative results. Finally, during an *in situ* experience at a 24 m deep site in the Adriatic Sea, Langlet et al., (2013, 2014) observed a decreased living foraminiferal density (labelled with CTG), but also found that almost all species survived after 10 months of anoxia with co-occurrence of $H_2S$ in the water column and sediment. However, the duration of sulphidic

conditions was estimated to last for several weeks but could not be assessed precisely (Metzger et al., 2014).

In our study, at station 1, bottom waters were hypoxic in July 2012 and became anoxic in August (Fig. 11). Both in July and August, oxygen penetration into the sediment was null, whereas it was 0.7 mm in September. In all three months (July to September 2012), sulphidic conditions were observed very close to the sediment-water interface (1 mm or less, Fig. 11). In view of these results, the duration of anoxic and sulphidic conditions in the uppermost sediment layer can be estimated as one

to two months (in July and August, Fig. 11).

After the strong increase of foraminiferal densities in spring 2012, there is a strong decrease starting in July, leading to a near-absence of foraminifera in November (Fig. 11). The most probable cause of the strong decline of the foraminiferal community appears to be a prolonged presence of sulphides in the foraminiferal microhabitat. However, the fact that foraminiferal abundances reached almost zero only in November (two months after the last stage of sulphidic conditions in the upper

sediment, in September) suggests that the presence of $H_2S$ did not cause instantaneous mortality, but that the disappearance of the foraminiferal community was a delayed response, probably caused by inhibited reproduction and, eventually, increased mortality.

Such a time lag between a drop or an increase in abundances in response to changes in environmental parameters affecting reproduction and/or growth of foraminifera was already suggested by Duijnstee et al. (2004). The authors highlighted that the

dynamics of some foraminiferal species showed higher correlation with measured environmental parameters (e.g., oxygenation or temperature) when a time lag of about three months was applied.

For 2011 at the same station, no pore-water $O_2$ and $H_2S$ measurements are available. However, severe hypoxia was observed in the bottom waters from May to August, with anoxia in June 2011 (Fig. 11). We therefore assume that like in 2012, anoxic and probably co-occurring sulphidic conditions were responsible for the very low standing stocks in August and November

2011 and January 2012.

Our observations confirm the suggestion of Moodley et al. (1998a) that foraminifera cannot withstand a prolonged presence of $H_2S$ in their habitat. Inhibition of reproduction has earlier been suggested as a response to hypoxic/short anoxic (Geslin et al., 2014) and sulphidic conditions (Moodley et al., 1998b).

After the 2011 hypoxia/anoxia, standing stocks at station 1 only started to increase in March 2012, indicating a very long

recovery time (about 6 months) of the foraminiferal faunas after a temporary near-extinction due to anoxic and sulphidic conditions.



This confirms observations of relatively long recovery times in the literature (e.g. Alve, 1995, 1999; Gustafsson and Nordberg, 2000; Hess et al., 2005). For instance, Gustafsson & Nordberg (1999) showed that in the Koljö Fjord, at comparable water depths, foraminiferal populations responded with increased densities only three months after a renewal of sea-floor
oxygenation following hypoxic conditions in bottom water. However, in that case, the disappearance of the foraminiferal was not nearly complete, as in our study.

At station 2, in 2012, hypoxia was only observed in August, when the OPD was zero, and sulphidic conditions were observed in the superficial sediment (i.e. from 0.4 mm downwards, Fig. 12). Both in July and September, oxygen penetrated more than one millimetre into the sediment. However, free $H_2S$ was still detected at about two millimetres depth in the sediment. Although
the sampling plan does not allow us to be very precise about the duration of anoxic and sulphidic conditions, we can estimate their duration to be 1 month or less (Fig. 12).

Foraminiferal abundances showed a strong decrease in October and November 2012, two months after the presence of anoxic and sulphidic conditions in the topmost part of the sediment (Fig. 12). Like at station 1, this temporal offset lag between the presence of anoxia/sulphidic conditions at station 2 (in August) and the strong decrease of faunal densities may be explained
as a delayed response, mainly due to inhibited reproduction during the anoxic/sulphidic event. If true, in the months after the presence of $H_2S$ in the uppermost sediment, the mortality of adults did not strongly increase, but they were no longer replaced (in the >125 µm fraction) by growing juveniles, because reproduction was interrupted when $H_2S$ was present in the foraminiferal microhabitat. Renewed recruitment after the last stage of sulphidic conditions somewhere in September would then explain why the faunal density in the >125 µm fraction increased again in December 2012.

In 2011, at station 2, bottom waters oscillated between hypoxic and oxic conditions between May and August (Fig. 12). Although we have no measurements of $H_2S$ in the pore waters for this year (i.e. like at station 1), it seems probable that bottom water hypoxia was accompanied by the presence of free $H_2S$ very close to the sediment surface, strongly affecting the foraminiferal communities. If we assume that, like in 2012, rich foraminiferal faunas were present in spring 2011 at both stations, the low faunal densities observed in August and November 2011 could suggest that also in 2011, foraminifera show
a delayed response to sulphidic conditions.

It is interesting to note that the foraminiferal densities observed at station 2 in August 2011 were lower than in July or September 2012. This might be attributable to the repetition of short hypoxic events in the bottom water between May and August 2011 (probably associated with anoxia and maybe $H_2S$ in the uppermost part of the sediment), which possibly affected the foraminiferal community more substantially than in 2012, when a hypoxic event was only recorded in August.

The important decrease of total standing stocks at station 2 in October and November 2012, (Fig. 12) suggests that in spite of the shorter duration of anoxia and sulphide conditions (compared to station 1; one month or less compared to one to two months), the foraminiferal faunas had still been strongly affected. However, at station 2, foraminiferal abundances increased again in December 2012, suggesting a recovery time of about two months, much shorter than at station 1, where standing stocks in the >125 µm fraction only increased 6 months after the presence of anoxia and free sulphides.





Summarising, the foraminiferal communities of both stations 1 and 2 seem to be strongly impacted by the anoxic and sulphidic conditions in the uppermost part of the sediment developing in late summer/early autumn. However, at station 1, where anoxic and sulphidic conditions lasted for one to two months, the response is much stronger, leading ultimately (in November) to almost complete disappearance of the foraminiferal fauna. The delayed response at both stations shows that mortality has not been instantaneous, and suggests that the decreasing standing stocks are the result of inhibited reproduction, and eventually,

increased mortality.

Recovery is much faster at station 2 (about two months) than at station 1 (about six months), probably because at station 1 (in contrast to station 2) the foraminiferal extinction was nearly complete, and the site had to be recolonised (e.g. by nearby sites or by the remaining few individuals) after reoxygenation of the sediment. At station 2, a reduced but significant foraminiferal community remained present, explaining the faster recovery.

**4.4 Species-specific response to environmental conditions in Lake Grevelingen**

As species determinations are increasingly based on genetic evidence and studies based only on morphological identification may suffer of taxonomic bias (Pawlowski and Holzmann, 2014), the comparison with earlier studies is difficult. Therefore, we have restricted our comparisons to studies with relatively similar environmental conditions and whenever possible, with clear SEM images.

The assemblages of Lake Grevelingen were dominated by *E. selseyense*, *E. magellanicum* and *Ammonia* sp. T6 at station 1 and the same three species plus *T. inflata* at station 2. *Elphidium selseyense*, *E. magellanicum* and *Ammonia* sp. T6 are very commonly found in coastal intertidal mudflats and/or other shallow water environments (e.g. Gustafsson and Nordberg, 1999, 2000; Langer and Leppig, 2000; Murray and Alve, 2000; Armynot du Châtelet et al., 2011; Schweizer et al., 2011; Saad and Wade, 2016). *Trochammina inflata* is an estuarine species with a worldwide distribution, which is typically found in salt

marshes in the upper estuary (Debenay et al., 2006; Horton and Murray, 2007). However, other species of *Trochammina* are also commonly found in low DO environment (Gupta, 2007).

To our knowledge, all earlier studies show that the foraminiferal response to hypoxia/anoxia is species-specific (e.g. Bernhard and Alve, 1996; Ernst et al., 2005; Bouchet et al., 2007; Geslin et al., 2014; Langlet et al., 2014). However, these species-specific responses generally follow the same scheme (usually decrease in density, reduction of growth and/or reproduction),

with different response intensities. Duijnstee et al. (2005) suggested that an oxic stress led to an increased mortality and an inhibited growth and reproduction. The suggestion of inhibited growth is supported by LeKieffre et al. (2017) who observed that *Ammonia tepida* showed minimal or no growth under anoxia. Conversely, Geslin et al. (2014) and Nardelli et al. (2014) suggested that reproduction was strongly reduced, but growth would not be affected by hypoxic and/or short anoxic events. Additionally, it is known that under low oxygen conditions, many species are able to shift to an anaerobic metabolism, such

as denitrification (Risgaard-Petersen et al., 2006; Piña-Ochoa et al., 2010a), or by entering into a state of dormancy (Ross and Hallock, 2016; LeKieffre et al., 2017).



Our study of the size distribution of *E. selseyense* and *Ammonia* sp. T6 shows an absence of clear cohorts, suggesting that reproduction takes place throughout the year. Continuous reproduction during the year has been described earlier for different foraminiferal genera, such as *Elphidium, Ammonia, Haynesina, Nonion* and *Trochammina* (e.g. Jones and Ross, 1979; Murray, 1983; Cearreta, 1988; Murray, 1992; Basson and Murray, 1995; Gustafsson and Nordberg, 1999; Murray and Alve, 2000). However, for *Ammonia* sp. T6, a rapid increase of overall test size between March and May could be indicative of a period of increased growth in spring (Fig. 7), possibly in response to a food input following phytoplankton blooms in April–May (Fig. 10, Hagens et al., 2015).

The comparison of the faunal dynamics at the two investigated stations and the different seasonal patterns of the major species allow us to draw some conclusions about interspecific differences in the response to seasonal anoxic and sulphidic conditions. First, there is a clear faunal difference between the two stations. Station 1 is dominated by *E. selseyense* and *E. magellanicum* while at station 2, these two taxa are accompanied by *Ammonia* sp. T6 and *T. inflata*. The latter species is almost absent at station 1, whereas *Ammonia* sp. T6 is present with very moderate densities. At first view, this would suggest that the two *Elphidium* species have a greater tolerance to the seasonal anoxic and sulphidic conditions.

Furthermore, it is interesting to note that the temporal evolution of standing stocks at station 1 is different between the two *Elphidium* species. *Elphidium magellanicum* shows a strong drop in absolute density in July 2012, at the onset of H$_2$S presence in the uppermost part of the sediment, whereas the diminution of *E. selseyense* is more progressive and the species disappears almost completely only in November (Fig. 4). This strongly suggests that *E. magellanicum* is more affected by increased mortality than *E. selseyense* due to the combined effects of anoxic and sulphidic conditions. This conclusion is confirmed by the patterns observed at station 2, where the drop in standing stock in October–November is also more drastic in *E. magellanicum* than in *E. selseyense* (Fig. 5).

At station 2, it is also remarkable to see that both *Ammonia* sp. T6 and *T. inflata* show maximum densities in winter (January–March), contrasting with the two *Elphidium* species, which have their density maxima later in the year (May–September). This could be explained by a difference in preferential food sources.

Foraminifera exhibit a large range of feeding strategies, some are selective feeders (Muller, 1975; Suhr et al., 2003; Chronopoulou et al., 2019). Hagens et al. (2015) reported that in Lake Grevelingen the phytoplankton composition was different between spring and summer 2012. In April–May, the phytoplankton bloom was mainly composed of the haptophyte *Phaeocystis globosa*, whereas it was dominated by the dinoflagellate *Prorocentrum micans* in July. *Elphidium* was reported to be able to feed on various food sources (e.g. diatoms, dinoflagellates, green algae; Correia and Lee, 2002; Pillet et al., 2011). However, diatoms should be the major food source for kleptoplastic species (Bernhard and Bowser, 1999), such as *E. selseyense* (Jauffrais et al., 2018). *Ammonia* sp. seems able to feed on very diverse food sources including microalgae, diatoms, bacteria or even metazoans (Lee et al., 1969; Moodley et al., 2000; Dupuy et al., 2010; Jauffrais et al., 2016; Chronopoulou et al., 2019). Recently, Chronopoulou et al. (2019) showed different feeding preferences for *Ammonia* sp. T6 and *E. selseyense* in intertidal environments in the Dutch Wadden Sea. Although diatoms are harvested by both species (but in different proportions), dinoflagellates were consumed by *E. selseyense* but not by *Ammonia* sp. T6, which feeds also on metazoans by





active predation (see also Dupuy et al., 2010). Jauffrais et al. (2018) showed that *E. selseyense* is able to sequester chloroplasts from ingested diatoms, and to keep them active for several days to weeks. These active chloroplasts could serve as an alternative source of oxygen and/or food through photosynthesis (if the amount of light is sufficient as shown at 45 m depth in a fjord for *Stainforthia fusiformis*, Bernhard and Alve, 1996) or another metabolic pathway (Jauffrais et al., 2019), and thereby

increase the capability of this species to survive anoxic events. Although sequestration of chloroplasts was never shown in *E. magellanicum*, its abundant spinose ornamentation in the umbilical region and in the vicinity of the aperture (Fig. 3c–d) strongly suggests that this species is capable to sequester chloroplasts as well (Bernhard and Bowser, 1999; Austin et al., 2005), which could partly explain its resilience to anoxia and sulphidic conditions.

The drop in foraminiferal densities at station 2 in October–November, which we interpreted as a delayed response to sulphidic

conditions, is less strong for *Ammonia* sp. T6, suggesting that this species is less affected than the two *Elphidium* species. This does not agree with our earlier suggestion that *Elphidium* species would be more tolerant to anoxic and sulphidic conditions. An explanation for this apparent contradiction could be that food sources available in spring were more suitable for *E. selseyense and E. magellanicum* than for *Ammonia* sp. T6. At station 2, the decreasing densities of *Ammonia* sp. T6 between March and May 2012 may be due to a lack of recruitment, with a continuing size increase of the adult specimens (Fig. 7).

Conversely, *E. selseyense* (and *E. magellanicum*) would continue to reproduce in spring, leading to progressively increasing densities, and an absence of clearly defined cohorts with a high proportion of small sized specimens (Fig. 6).

These observations seem to indicate that at station 2, the difference in population dynamics between *Ammonia* sp. T6 and the two *Elphidium* species does not denote a large difference in tolerance to anoxia/sulphides, but rather a different adjustment of *Ammonia* sp. T6 and the two *Elphidium* species with respect to the seasonal cycle of food availability.

The very low densities of *Ammonia* sp. T6 at station 1 could then be explained by a recolonization starting in (late) winter, with only a few individuals present in January, at the end of the late autumn/early winter season with favourable food conditions for this taxon (as testified by the very strong density increase in January 2012 at station 2). Once a more abundant pioneer population was present (in early spring), food conditions were no longer favourable for *Ammonia* sp. T6, but were optimal for the two *Elphidium* species, explaining the strong dominance of the latter two species at station 1. If true, the lower densities of

*Ammonia* sp. T6 would not be due to a lower resistance to anoxia and free sulphides, but rather due to an unfavourable seasonal succession of food availability.

Previous studies already suggested that hypoxic/anoxic conditions coupled with increased food input from autumnal phytoplankton blooms (composed of diatoms and dinoflagellates) would favour the development of *E. magellanicum* (Gustafsson and Nordberg, 1999). The fact that also at station 2, this species was mainly observed between March and

September 2012 corroborates our conclusion of its dependence on a specific food regime.

Finally, encrusted forms of *E. magellanicum* were observed at both stations from May until the end of the year, but were absent in the samples of March 2012. The observation of abundant specimens covered by feeding cysts corroborates our conclusion that the surface water phytoplankton bloom in May 2012 (i.e. probably mainly *Phaeocystis globosa*) provided a food source particularly well suited to the nutritional preferences of this species.



## 5 Conclusion


In both stations investigated, foraminiferal communities are highly impacted by the combination of anoxia and H$_2$S in their habitat. The foraminiferal response varied depending on the duration of adverse conditions, and led to near extinction at station 1, were anoxic and sulphidic conditions were present for one to two months, compared to a drop in standing stocks at station 2, where these conditions lasted for one month or less. At both sites, foraminiferal communities showed a two months delay

in the response to anoxic and sulphidic conditions, suggesting that the presence of H$_2$S inhibited reproduction, whereas mortality was not necessarily increased. The duration of the subsequent recovery depended on the fact whether the foraminiferal community was almost extinct (station 1) or remained present with reduced effectives (station 2). In the former case, about six months was needed for faunal recovery, whereas in the latter case, it took only two months. We hypothesize that the dominance of *E. selseyense* and *E. magellanicum* at station 1 is not due to a lower tolerance to anoxic and sulphidic

conditions of *Ammonia* sp. T6, but is rather the consequence of a different adjustment between the two *Elphidium* species and *Ammonia* sp. T6 with respect to seasonal cycle of food availability.

**Data availability**

Raw data are available in Supplementary Material.

**Author contributions**

J.R.: generated the size distribution data. B.R. and D.L. picked the foraminifera. D.S.: provided geochemical data. All authors contributed to the writing of the manuscript.

**Competing interests**

The authors declare that they have no conflict of interest.

**Acknowledgements**

We are very grateful to Sandra Langezaal for inviting us to study the fascinating environments of the Grevelingenmeer. We acknowledge the support of P. van Rijswijk, M. Hagens, A. Tramper, and the crew of the R/V Luctor (P. Coomans and M. Kristalijn) during the sampling campaigns. We acknowledge Jassin Petersen for his help with recovering some of the environmental data. This study profited from funding of Rijkswaterstaat and of the CNRS program CYBER-LEFE (project AMTEP).



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





**Table 1:** **Sampling dates for stations 1 and 2. x = one core investigated, o = no core investigated**

| Year | Month | Day | Station 1 | Station 2 |
|------|-------|-----|-----------|-----------|
| 2011 | August | 22 | x x | x x |
| 2011 | November | 15 | x x | x x |
| 2012 | January | 23 | x x | x x |
| 2012 | March | 12 | x x | x x |
| 2012 | May | 30 | x x | x x |
| 2012 | July | 24 | x x | x x |
| 2012 | September | 20 | x x | x x |
| 2012 | October | 18 | o | x x |
| 2012 | November | 2 | x x | x x |
| 2012 | December | 3 | o | x x |





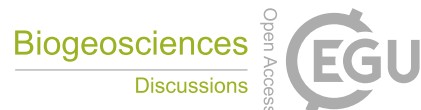


**Table 2: Mean living foraminiferal abundances (ind. 10 cm$^{-3}$) and relative abundances (between brackets) of the dominant species and total assemblage in 2011 and 2012 for both stations 1 (top) and 2 (bottom).**

## STATION 1

| Year | Month | *Elphidium selseyense* | *Ammonia* sp. T6 | *Elphidium magellanicum* | *Trochammina inflata* | Others | Total assemblage |
|------|-------|------------------------|------------------|--------------------------|-----------------------|--------|------------------|
| 2011 | August | 1.2 (36.8%) | 1.2 (36.8%) | 0 (0%) | 0 (0%) | 0.9 (26.3%) | 3.4 (100%) |
| 2011 | November | 0.5 (50%) | 0.4 (33.3%) | 0 (0%) | 0 (0%) | 0.2 (16.7%) | 1.1 (100%) |
| 2012 | January | 5.1 (44.6%) | 3.2 (27.7%) | 0.2 (1.5%) | 1.2 (10.8%) | 1.8 (15.4%) | 11.5 (100%) |
| 2012 | March | 23.9 (38.5%) | 12.9 (20.8%) | 21.6 (34.8%) | 1.4 (2.3%) | 2.3 (3.7%) | 62.1 (100%) |
| 2012 | May | 336.5 (74.8%) | 9.2 (2%) | 96.4 (21.4%) | 1.8 (0.4%) | 6 (1.3%) | 449.9 (100%) |
| 2012 | July | 162 (90.2%) | 10.3 (5.7%) | 3.7 (2.1%) | 0 (0%) | 3.5 (2%) | 179.5 (100%) |
| 2012 | September | 29.7 (87.5%) | 2.3 (6.8%) | 0 (0%) | 0.4 (1%) | 1.6 (4.7%) | 34 (100%) |
| 2012 | November | 1.1 (66.7%) | 0.4 (22.2%) | 0 (0%) | 0 (0%) | 0.2 (11.1%) | 1.6 (100%) |
| | Sum | 560 (75.4%) | 39.8 (5.4%) | 121.8 (16.4%) | 4.8 (0.6%) | 16.4 (2.2%) | 742.9 (100%) |

## STATION 2

| Year | Month | *Elphidium selseyense* | *Ammonia* sp. T6 | *Elphidium magellanicum* | *Trochammina inflata* | Others | Total assemblage |
|------|-------|------------------------|------------------|--------------------------|-----------------------|--------|------------------|
| 2011 | August | 74.8 (43%) | 82.1 (47.2%) | 0 (0%) | 14.7 (8.4%) | 2.5 (1.4%) | 174 (100%) |
| 2011 | November | 52.3 (40.7%) | 60.8 (47.3%) | 0 (0%) | 11.8 (9.2%) | 3.7 (2.9%) | 128.7 (100%) |
| 2012 | January | 161.8 (30.9%) | 226.2 (43.2%) | 0.9 (0.2%) | 121.5 (23.2%) | 13.3 (2.5%) | 523.6 (100%) |
| 2012 | March | 214.7 (38.2%) | 214 (38.1%) | 48.8 (8.7%) | 75 (13.3%) | 9.9 (1.8%) | 562.3 (100%) |
| 2012 | May | 288.2 (47.7%) | 147.1 (24.3%) | 116 (19.2%) | 36.1 (6%) | 17.3 (2.9%) | 604.8 (100%) |
| 2012 | July | 282.6 (53.2%) | 158.4 (29.8%) | 37.8 (7.1%) | 31.5 (5.9%) | 21.2 (4%) | 531.6 (100%) |
| 2012 | September | 365.5 (64.4%) | 102.4 (18%) | 72 (12.7%) | 16.1 (2.8%) | 11.5 (2%) | 567.5 (100%) |
| 2012 | October | 98.7 (46.7%) | 99 (46.8%) | 1.8 (0.8%) | 7.4 (3.5%) | 4.6 (2.2%) | 206.9 (100%) |
| 2012 | November | 30.9 (34%) | 48.1 (52.8%) | 4.1 (4.5%) | 3.7 (4.1%) | 4.2 (4.7%) | 91.1 (100%) |
| 2012 | December | 252.2 (66.7%) | 78 (20.6%) | 25.5 (6.7%) | 12.7 (3.4%) | 9.5 (2.5%) | 368.4 (100%) |
| | Sum | 1821.8 (48.3%) | 1216.1 (32.2%) | 306.8 (8.1%) | 330.5 (8.8%) | 83.6 (2.6%) | 3758.9 (100%) |





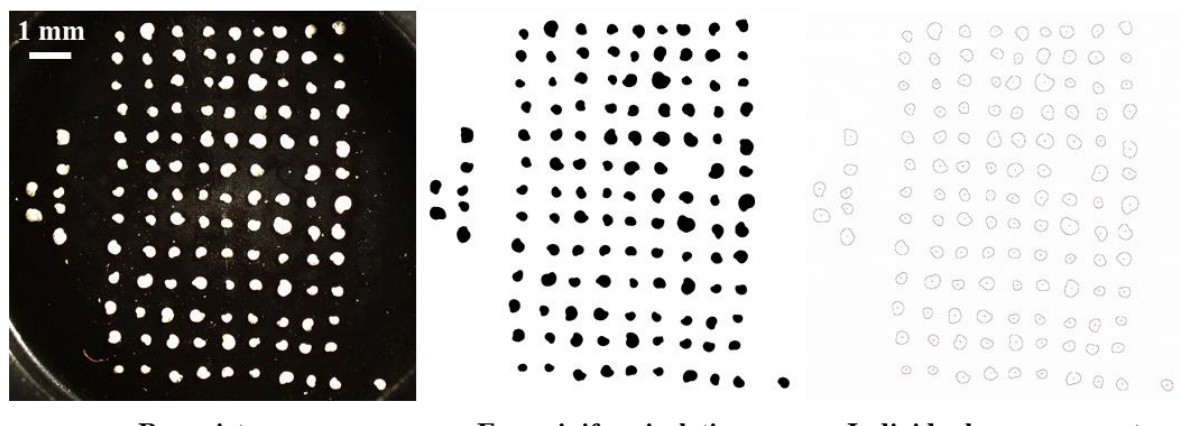


**Figure 1:** **This figure shows the different steps of the numerical treatment of each image. The left figure shows the untreated image, the middle figure presents the next step, when all individual foraminifera are depicted. Finally, the figure on the right shows the individual foraminiferal outlines which were measured.**



**Figure 2:** The grey bars represent the living foraminiferal abundances for the two replicates. The mean abundances (diamonds) and standard deviations (black error bars) were calculated for the two replicates for stations 1 (34 m depth, top panel) and 2 (23 m depth, bottom panel). All abundance values are for the 0–1 cm layer and were standardised to 10 cm³. Months for which foraminiferal communities were investigated are indicated in bold.







**Figure 3:** **SEM images of *Elphidium selseyense* in lateral (a) and peripheral (b) view, *Elphidium magellanicum* in lateral (c) and peripheral (d) view, *Ammonia* sp. T6 in spiral (e), peripheral (f) and umbilical (g) view, and *Trochammina inflata* in spiral (h), peripheral (i) and umbilical (j) view. All scale bars are 50 μm.**





**Figure 4:** The bars represent the living foraminiferal abundances for the two replicates for *Elphidium selseyense* (blue), *Elphidium magellanicum* (green), *Ammonia* sp. T6 (orange) and *Trochammina inflata* (yellow) at station 1 in 2011 and 2012. The mean abundances (diamonds) and standard deviations (black error bars) were calculated for the two replicates. All abundances values are for 0–1cm layer and were standardised to 10 cm³. Months where foraminiferal communities were investigated are indicated in bold. Scales were chosen in order to facilitate comparison with station 2.





**Station 2**



**Figure 5:** The bars represent the living foraminiferal abundances for the two replicates for *Elphidium selseyense* (blue), *Elphidium magellanicum* (green), *Ammonia* sp. T6 (orange) and *Trochammina inflata* (yellow) at station 2 in 2011 and 2012. The mean abundances (diamonds) and standard deviations (black error bars) were calculated for the two replicates. All abundances values are for 0–1cm layer and were standardised to 10 cm³. Months where foraminiferal communities were investigated are indicated in bold. Scales were chosen in order to facilitate comparison with station 2.





**Figure 6:** size distribution (maximum diameter for each individual in µm) of *Elphidium selseyense* for stations 1 (left) and 2 (right) in 2012. For each month, the number of individuals (n), the mode and the number of individuals associated to the mode (between brackets) are indicated in black. The medians are indicated by the red bars in each panel.







**Figure 7:** size distribution (maximum diameter for each individual in µm) of *Ammonia* sp. T6 for stations 1 (left) and 2 (right) in 2012. For each month, the number of individuals (n), the mode and the number of individuals associated to the mode (between brackets) are indicated in black. The medians are indicated by the red bars in each panel.



Figure 8: SEM images of (a) fully encrusted specimen, (b) partially encrusted specimen, (c) crushed encrusted specimen of *Elphidium magellanicum*. Note the thinness of the crust and the spinose structures on (d) and (e).





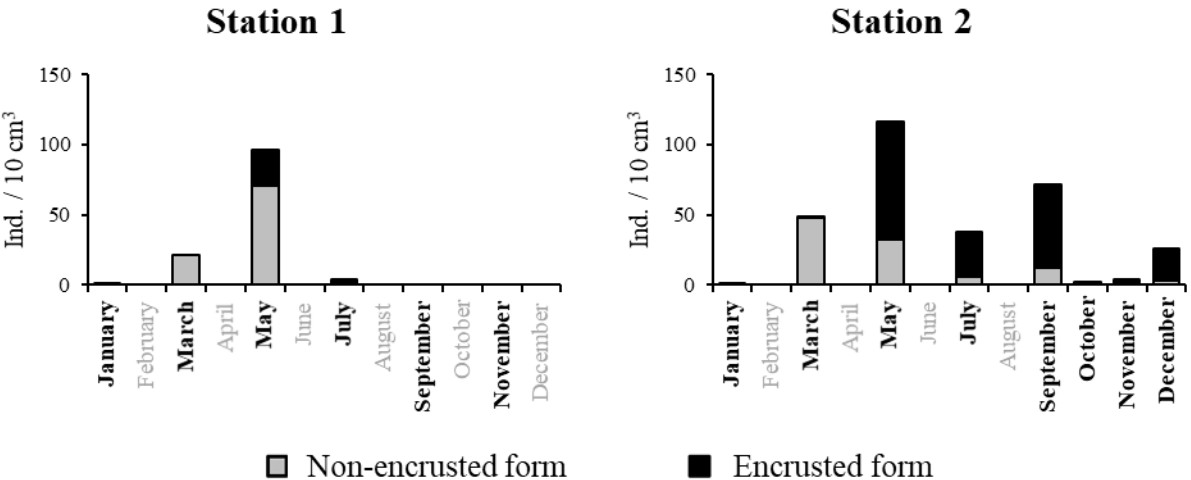

**Figure 9:** Mean abundances (ind. 10 cm$^{-3}$) of non-encrusted (grey) and encrusted forms (black) of *Elphidium magellanicum* in 2012, at station, 1 (left) and 2 (right). Investigated months are indicated in bold.

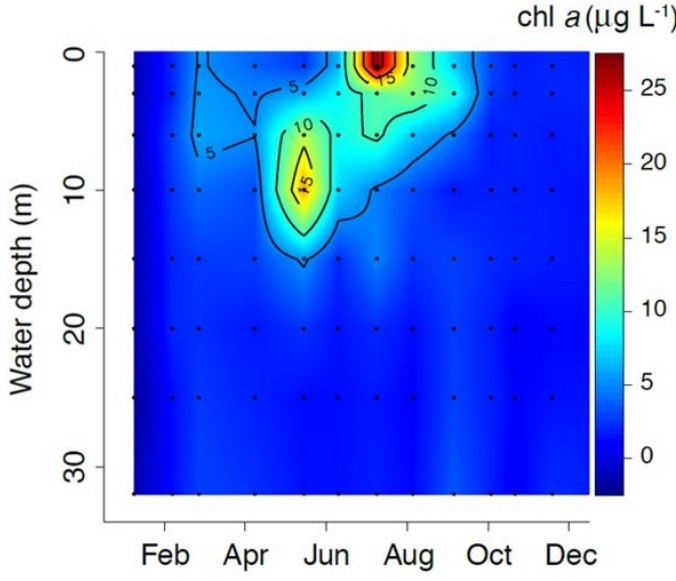


Figure 10: Monthly Chl *a* concentrations (µg L$^{-1}$) in the water column in Den Osse Basin in 2012 – From Hagens et al. (2015).



**Figure 11: The top panel represents bottom water oxygen concentrations (µmol L⁻¹) in 2011 and 2012 at station 1, from Donders et al. (2012) and Seitaj et al. (2017). The grey horizontal dotted line indicates the hypoxia limit (63 µmol L⁻¹). The middle panel represents the depth (in mm) distribution of the oxic (blue), suboxic (orange) and sulphidic (black) zones within the sediment in 2012, from Sulu-Gambari et al. (2015). The bottom panel shows the total living foraminiferal abundances for both replicates (grey bars), mean abundances (diamonds) and standard deviations (black error bars) calculated for the two replicates, for all investigated months (in bold) in 2011 and 2012.**



Figure 12: The top panel represents bottom water oxygen concentrations (µmol L⁻¹) in 2011 and 2012 at station 2, from Donders et al. (2012) and Seitaj et al. (2017). The grey horizontal dotted line indicates the hypoxia limit (63 µmol L⁻¹). The middle panel represents the depth (in mm) distribution of the oxic (blue), suboxic (orange) and sulphidic (black) zones within the sediment in 2012. The bottom panel shows the total living foraminiferal abundances for both replicates (grey bars), mean abundances (diamonds) and standard deviations (black error bars) calculated for the two replicates, for all investigated months (in bold) in 2011 and 2012.