# Peer review of "Foraminiferal community response to seasonal anoxia in Lake Grevelingen (the Netherlands)"

_Biogeosciences, 2019_

## Referee Comment (RC1) · Anonymous Referee #1 · 25 Oct 2019

**Review of the manuscript bg-2019-382.**

**General comments**

The manuscript entitled "Foraminiferal community response to seasonal anoxia in Lake Grevelingen (the Netherlands)" by Richirt et al. consists of a field study which aims to analyze the benthic foraminiferal community characteristics during 1.5 years, during when seasonal hypoxia/anoxia occurs together with presence of $H_2S$. The results are very interesting and will be useful for the community, and the figures are clear and informative. However, the paper is poorly structured, and some statements need to be taken more carefully. Therefore, I suggest the following revisions before publication in Biogeosciences.

A restructuration of the material and methods section seems necessary. I suggest to create a studied area section, separated from the material and methods, with a description of the lake and the Den Osse Basin. It would be especially interesting as, as the authors say in the introduction, "a large amount of environmental data is available" (Line 64). This section should include all the references cited in the second part of section 2.1 (which should then be deleted) and in section 4.2 from the discussion. This section 4.2 is a description of already published results, not a discussion of the results from this paper. I suggest moving this paragraph to the studied area section.

The CTG method and a comparison with the Rose Bengal method was published in 2006, which is thirteen years ago. Since then, many studies have successfully used CTG to label their samples. I do not think that this part of the discussion about why the authors have chosen this method is necessary. The whole section 4.1 is a repetition of what you say in the introduction and does not bring anything new, it should be removed.

The discussion needs to be restructured as well. The first two paragraphs of section 4.3 are a description of the literature, not a discussion. I would move them in the introduction section, and cite these references where they are relevant, later in the discussion. Similarly in section 4.4, the 3rd paragraph is literature description that should or go in the introduction and/or be included later when discussing the results of the paper. I strongly suggest to reconsider the whole structure of the section 4.4, which I find not easy to read in the current state.

The authors should be very careful about the reaction times they give in the discussion and conclusion. For example, line 329, the "two months after" cannot be assed for sure, as we do not have information about the fauna in October. Picking does take a lot of time, and I understand that picking all the months was not possible, but I recommend using more approximate times, especially for station 1. Moreover, for me at station 1 on Fig. 11, the foraminiferal response to $H_2S$ appears immediate, as their abundance is lower already in July. Could the authors explain?

In the results, a full paragraph is dedicated to encrusted forms, together with a full plate of pictures, and two detailed graphs. I suggest to strongly develop this part in the discussion, which now consist of 4 lines, to include the information from  the second paragraph of section 3.4 – and explanation given by these authors -, and the following references: Cedhagen 1996 (Phuket Marine Biological Center), and Heinz et al. 2005 (Marine Biology Research). Please also explain the current statements, do you suggest that the feeding cysts only get formed when *P. globosa* is blooming?

Please find below minor suggestions and text comments.

**Minor comments**

Abstract

Line 19: early diagenesis and organic matter recycling are mentioned here but never again in the paper. Please explain.

Line 30: This is in contradiction with your conclusion, is there not a "drop in standing stocks" for station 2?

Line 34-35: The two sentences are in contradiction, please rephrase.

Line 32: Replace "H2S" by "$H_2S$".

Introduction

Please shortly explain what are foraminifera, what are their place and role in these types of environment, and why you chose them for your study.

Line 43-46: This sentence is long and confusing, please rephrase.

Line 46: Could you give some examples of these consequences?

Line 50: I suggest to specify which ones of these references are field or culture studies, and to reconsider the sentence accordingly.

Line 52: Could you explain why/how anoxia and $H_2S$ are linked?

Lines 77-78: These references are already cited earlier. Please restructure.

Line 81: Please add references, even if they are "sparse".

Line 82: Is there not any previous foraminiferal studies in the lake itself?

Lines 82-92: Please shorten this part, the CTG method is well known already.

Lines 96-97: This belongs to the method section, please remove.

Line 100: example of these indices?

Material and Methods

The description of the lake is not a part of the method. See also my general comments.

Please specify that SEM pictures were taken for the four dominant species including encrusted specimens, with which microscope and where were they taken.

Lines 112-114: This paragraph should be moved to the field sampling section.

Line 114: I guess a map is available in the cited paper? Maybe you can precise it here?

Line 118: "similarly", by who?

Line 120: Please give more details about the sampling. I see in the acknowledgments that the r/v Luctor was involved. What kind of corer was used? How long where the cores? Were some environmental data taken at the same time?

Line 123: I know that CTG labelling happens on the field. But after that you talk about picking. As this is not a field sampling event, I would move this to the sample treatment section.

Line 127: Add "finally" before "investigated". See my comment about Table 1.

Line 133: "previous studies", where were they? Please add references.

Line 166: On which species was this done, and how many specimens were used?

Results

Line 178-179: Remove this sentence. You already explain it in the method, and the Figure 2 has a caption.

Line 179: Please check if you mean total or mean abundances here.

Line 183: I would be careful with the use of "early" and "late", talking about the seasons. July is not early summer. Line 433, March is not winter. Please check through the paper, maybe giving the months is the most accurate solution.

Lines 194-195: Please remove this sentence, you have already explained in the method.

Line 197: Replace "Fig.4" by "Figure 4". This sentence should be moved to the method section.

Lines 203-204: I suggest to remove this sentence, it does not bring anything new.

Line 206: Add "and Table 2" after Fig. 4.

Line 211: Remove "(fairly low)"

Line 213: We know that *T. inflata* was absent in 2011, as you said it line 205. Please rephrase. I think the way you described the results for the station 2 is clearer than for station 1. I suggest to also describe the station 1 species by species, instead of year by year.

Line 225: Please remove "Conversely to station 1", this is confusing here.

Lines 230-233: This should be moved to the methods section.

Line 234: Please add "(Fig. 6)" after "station 2".

Lines 256 and 258: same information, please modify.

Line 260: "Similar observations", where?

Line 260-262: This part should go to the discussion section.

Line 263: Please remove this sentence and cite the Figure 9 in the following sentence.

Discussion

I think the section 4.1 should be removed from the paper. See also my general comments.

The information given in the 4.2 section are not results from this paper, they are a description of the site citing already published papers. This should go in the studied area section. See also my general comments.

In section 4.3, the actual discussion starts on line 321. See also my general comments.

Line 337: Do you have information about why the 2011 hypoxia was so severe compared to the 2012 one?

Line 339: I know that this study only focus on living fauna, but it would have been interesting to check the dead fauna further down in the cores, to see if standing stocks were indeed higher before the 2011 severe hypoxia.

Line 345: Could you explain how you deduced these "6 months" of recovery? As the hypoxia event was much more severe in 2011, how could we know if the $H_2S$ stayed longer in the upper sediment compared to 2012, and thus how long it affected the fauna? Please explain.

Line 366: Please remove "(i.e. like station 1)".

Line 365: This paragraph and the following one are very similar. I suggest to merge them.

Line 379: We cannot be sure about that, as there are no available data. Please modify this statement.

Line 386: Remove "(in contrast to station 2)".

Line 387: "by the nearby sites", I thought the water circulation was weak in the lake? Is transportation then possible? Please check.

Lines 395-396: This sentence belongs to the results section, please remove.

Line 437: But no diatoms?

Line 438: Which *Elphidium*? Elphidiids?

Line 440-441: Remove this sentence, it is confusing there, and you talk about this aspect just after.

Line 470: What about *T. inflata*?

Lines 476-479: This part should be developed. See also my general comments.

Conclusion

I would add a short introductory sentence or add details to the first sentence, to quickly remind the reader what you did.

References

Biogeosciences is very careful with bibliography details. Please go through your references list: ~10 papers miss doi, some miss page range, etc.

Figures

Table 1: You say in the text that the sampling happened every month, but that you only analyzed specific months. Thus, it the title correct here?

Figure 1: In the caption, remove "This figure shows" in the first sentence, and add somewhere "for size measurement" as well as "ImageJ software".

Figure 2: I don't think "Total living assemblage" is necessary below Station 1 and Station 2. Instead, it would be better to have this as the vertical axis title, with the unit (ind. 10 cm$^{-3}$) into brackets. In the caption, replace "for which" by "where" to be consistent with other captions.

Figure 4: Vertical axis title?

Figure 5: Vertical axis title? Also, I guess you mean "station 1" in the last sentence. Please check the months in bold.

Figure 9: Vertical axis title? It would be informative to have the percentage of encrusted specimens on top of each bar.

Figure 11: You use the word "suboxic" in the caption, it's not coherent with the rest of the paper. Please check the months in bold.

Figure 12: "Figure 12".

I hope my comments will be taken by the authors in a spirit of constructive criticism with only intention to further improve their manuscript.

Sincerely, Laurie M. Charrieau

---

## Referee Comment (RC2) · Anonymous Referee #2 · 2 Nov 2019

Comments to the Author

Manuscript ID: bg-2019-382

The manuscript of Richirt and coauthors on "Foraminiferal community response to seasonal anoxia in Lake Grevelingen (the Netherlands)" represents the assemblage fluctuation of benthic community in response to hypoxic/anoxic environments. These analyses are important to understand the foraminiferal tolerance to hypoxia/anoxia and hydrogen sulfide, and to understand life histories under the extreme environments. However, the structure of manuscript is very poor and experimental design and data validations are problematic. Therefore, I strongly suggest reconstructing throughout the manuscript, and also data validation is needed.

The biggest problem is that authors only used specimens of 125μm or more. Juvenile specimens have the size smaller than 125μm. If you are looking at population dynamics, you must deal with juvenile specimens.

In the methods section, the authors should explain more detailed procedures. I also found several methods sentences in the result and also in the discussion. Also, in the section 3.4, the authors described methods, although this paragraph is in the results section. These explanations should be move to appropriate section.

In the first section of the discussion (4.1), the author described both advantages and disadvantages of both CTG methods and Rose Bengal staining respectively. However, the CTG method has already described in Bernhard et al. (2006), and therefore it is not necessary to explain in detail.

In the Section 4.2, I strongly suggest that the author should describe environmental setting of the sampling points. However, these descriptions must be explained in the beginning of this article. The authors also should explain vertical profile of oxygen in the water column and in the sediment in the "environmental settings of Den Osse Basin" section. This information can help readers to understand the habitat where foraminifera live in.

In the section 4.3, I cannot understand what you want to discuss about. The authors referred (quoted) about previous studies in the first two paragraphs. The authors should move these paragraphs to the introduction, Ah...you would like to discuss relationship between sulfidic condition and foraminiferal assemblages? The discussion starts from line 321...
I strongly recommend to make clear and re-structure throughout the manuscript.

Other comments.

Line 34, "Elphidium selseyense and Elphidium magellanicum are much less affected by anoxia and free H2S than Ammonia sp. T6"

Is the light reaching the lake bottom? Is it not necessary to consider the photosynthesis of kleptoplasts? Ammonia T6 has a nitrate pool in the cell. Nomaki et al. (2014, Limnol. Oceanogr., 59, 1879–1888) points out that this species potentially use an anaerobic respiration.

Line 47-, "Benthic faunas are strongly impacted by these events (Diaz and Rosenberg, 1995) although the meiofauna, especially foraminifera, appears to be less sensitive to low Dissolved Oxygen (DO) concentrations than the macrofauna"

Virgulinella, Bulimina, etc. may be sensitive to anoxic environments. Cannariato et al (1999, Geology, 27, 63-66) has analyzed community changes over the last 60,000 years at Santa Barbara Basin. As a result, low-oxygen torelant species are clearly replaced. Bolivina tumida, Buliminella tenuata and Globobulimina auriculata are low oxygen torelant species (dysoxic species). Interestingly, the response to hypoxia varies from species to species. Buliminella tenuata increase at the beginning of dysoxic. On the other hand, Bolivina tumida increases toward to the end of the dysoxic period.
Bolivina tumida has symbiotic microbes in its cells. Bolivina pacifica, Uvigerina peregrina, and Loxostomum pseudoberyichi retain microbes outside (in the pore) (Bernhard et al. 2018, Mar. Micropal. 138, 33-45). Based on these phenomena, it is expected that the response pattern to anoxia will differ depending on the symbiotic mode. The authors should explain/discuss this phenomenon in introduction and discussions.

Line53-, "Neutral molecular H2S can diffuse through cellular membranes and inhibits the functioning of cytochrome c oxydase (a mitochondrial enzyme involved in ATP production), finally inhibiting an aerobic respiration (Nicholls and Kim, 1982; Khan et al., 1990; Dorman et al., 2002)."

What do you think about an anaerobic respiration? The authors should explain about an anaerobic repiration.

Line 89, "In order to avoid this problem, we used CellTracker™ Green (CTG) to recognise living foraminifera. CTG is a fluorescent probe which marks only living individuals with cytoplasmic (i.e. enzymatic) metabolic activity (Bernhard et al., 2006)"

This method is not new. The authors should only mention that CTG staining was used to distinguish live benthic foraminifera populations.

Line 115-, "Measurements of oxygen concentrations in the bottom water (1 m above the sediment-water interface using a CTD) for 2011 are from Donders et al. (2012), whereas the 2012 data are from Hagens et al. (2015) and Seitaj et al (2017). Oxygen Penetration Depth (OPD) and depth of free H2 S detection were determined using O2 and H2S microsensors by Seitaj et al., (2015) for station 1, and the data for station 2 were acquired similarly (Supplementary Table 1)."

The authors should explain about environmental settings both station 1 and 2 in the beginning. This information can help reader to understand faunal assemblage changes (and population dynamics). This information is in the end of this manuscript.

Line 121-, "The uppermost centimetre of each core was labelled with CellTracker™ Green CMFDA (CTG, 5-chloromethylfluorescein diacetate, final concentration of 1μM following Bernhard et al., 2006) and fixed in 5 % sodium borate buffered formalin after 24 h of incubation."

Where did you done this experiment? What kind of tools did you use? Did you sliced top 1cm and then put in the petri dish or some other container for CTG incubation? or jut put CTG directly onto the top of core? Need detailed experimental procedures.

Line 129, "125μm"

As the authors mention about juvenile specimens, it is important point. Juvenile specimens have smaller than 125μm in size in many cases. If you are looking at population dynamics, you should deal with juvenile specimens. For this reason, it is difficult to see when the juvenile specimens have been reproduced.

Line 145, "Supplementary Figure 1 shows…"

I found there are two types in these specimens. Specimens #145 and 152 have a larger proloculus than specimens #147 and #155. In my opinion, these differences in morphology correspond to different generations, megarospheric and microspheric. It is important points to find these generations to understand population dynamics. I strongly recommend to check which generations are abundant in each month.

Line 145, "the penultimate chamber"

Are there any differences in the pore size for each month (season)?

Line 165, "population dynamics"

Need juvenile specimens for analyze.

Line 169, "Fig. 1"

I think the authors should explain much more detail in this paragraph. Detailed procedures were written in figire1 caption!

Line 175, ">125 μm, our analysis mainly concerns adult specimens, anddoes not include juveniles. This limitation should be kept in mind when interpreting the results"

If the authors discuss about population dynamics, it is necessary to check juveniles.

Section3.1
Any statistical analyses?

Line 185, "very low in Janually"

I strongly suggest that the author should describe environmental setting of the sampling points. However, these descriptions must be explained in the beginning of this article. The authors also should explain vertical profile of oxygen in the water column and in the sediment in the "environmental settings of Den Osse Basin" section. This information can help readers to understand the habitat where foraminifera live in.

Line 193, section 3.2

It is better to explain one by one. The authors should explain about station 1 and then explain about station 2.

Line 221-, "then progressively decreased until the end of 2012 (= 48.1 ± 26) in November 2012). Trochammina inflata showed a similar pattern as Ammonia sp. T6"

It is necessary to indicate statistical analyses. Statistically significant?

Line 237, "of larger individuals (>400 μm)"

Are there any ecological meanings?

Line 239-, "The low number of Ammonia sp. T6 individuals at station 1 does not allow us to draw any firm conclusion concerning the size distribution at this station"

In the result section, the authors should describe "results" in detail. For example, there are several large sized individuals in May, simultaneously 200~250 μm-sized individuals are there.
How about propagules? Alve and Goldstein (2002, Journal of Micropaleontology, 21, 95-96; 2003, Limnology and Oceanography, 28, 2163-2170) discussed about propagules in their literatures.

Line 243, "but started to diminish in December"

Are there any data? Please provide.

Line 244, "decrease of the median to 339 μm"

Ammonia has two generations, asexual and sexual phases. These two generations are commonly found in spring and autumn. The authors have to think about the life cycle of foraminifera.

line 245-, "Our tentative to distinguish cohorts by using a deconvolution method to separate the total size distributions into a sum of Gaussian curves was not conclusive"

Please indicate in the methods section.

Line 246-251,

It is not a result.
If the goal is to evaluate foraminiferal behavior in an anaerobic environment, an experimental desing that analyzes small individuals should be considered. Objective 2 cannot be achieved.

Line 255, "thin (Fig. 8c– e) and rather coarse"

Are there any data?
To explain how it differs from the normal case, the authors should show the data.

Line 257-, "Because the crust stayed cohesive after exposition to 0.1 M of EDTA (EthyleneDiamineTetraacetic Acid) diluted in 0.1 M cacodylate buffer (acting as a carbonate chelator)"

This sentence should be moved to the methods section.

Line 259-,
This sentence should be move to the discussion section.

Line 269-281,
It is not necessary to explain detailed about disadvantages of Rose Bengal staining method and advantages of CellTracker Green.
Yes, the CellTracker Green labeling is suitable and reliable method to identify live specimens. However, incubation is required for the CTG method. I think this method includes some artifacts. During the staining, samples were transferred to petri dishes or bottles for 24 hours. The specimens were exposed different environmental condition from their habitat.
This paragraph can be more shorten. Because this method was already described in Bernhard et al (2006), so the authors do not need a detailed description of this method.

Line 291, Fig. 10
You can omit this figure. Because, this is not your data. You can mark the timing of blooming on your figures 11 and 12.

Line 328-,
In the case of symbiontic bacteria-bearing foraminifera, oxic condition is not suitable. Because symbiotic bacteria cannot consume hydrogen sulfide, methane or nitrate in an oxic

condition, and the host foraminifera cannot use organic matter and/or anaerobic respiration from microbes.

Line 368-
There is little data in 2011. This sentence is overstatement.
At both stations 1 and 2, low oxygen was observed from May to August. This situation is totally different from 2012. This characteristic situation will affect next year's (2012) assemblages.

Line 381-, "leading ultimately (in November) to almost complete disappearance of the foraminiferal fauna."

I'm worried about incubation time (duration) for CTG staining. For example, oxygen penetration depth is about 4mm in October at station 1, but sulfide layers still existed in the deep layer below 4mm. When the authors used top 1cm of the sediment for incubation, sulfidic conditions will be constructed in the experimental bottle (or other gear). For this reason, when living specimens still exist in top 4mm in October, sulfidic conditions may affect living ones. However, the authors did not explain detailed procedures of CTG staining methods. Long time exposure of sulfidic condition may affect living specimens. How did you evaluate for this effect in your experiment?

Line 384-, "inhibited reproduction, and eventually, increased mortality"

Need juvenile data.

Line 390, Section 4.4
It is not appropriate section title. Need improvement.

This section includes many topics related to environmental characteristics and food availability for foraminiferal responses. The authors should rearrange and clarify what authors want to discuss. This paragraph also includes the results. Need reconstruction.

Line 391-, 1st paragraph

Is this a topic sentence in this section? I think this information should be move to the Materials & Methods section. This section is also long and confusing. The authors have to reconstruct.

Line 413, "take place throughout the year"

Are there any evidences that reproduction took place throughout the year? The authors should describe detailed results in the Result section. There exist relatively small-sized specimens that increased in May and September-October-November. In my opinion, it looks reproduction occurred twice in 2012. However, it is difficult conclude that there are no three or four chambered juveniles.

---

## Author Comment (AC1) · 2 Dec 2019

**GENERAL ANSWER**

The two referees asked for an improvement of the structure of the manuscript, especially concerning the discussion section 4.4 of the original submission. We agree with the two referees that the structure of submitted manuscript could be improved. Consequently, we reorganised the manuscript following the referees' comments and suggestions.

List of corrections made to improve the structure of the manuscript highlighted by the two referees:

1- The whole section about size distribution (section 3.3 in the original submission) was moved to the supplementary material. Since the data do not allow us to make conclusive observations about the foraminiferal population dynamics, we now use the size data only as an additional argument corroborating our hypothesis of interspecific differences in preferred food sources (last section of the discussion). We state very clearly that these results should be considered with care in the Material and Methods (2.5) section.

2- Section 4.4 of the discussion in the original submission (now 4.2) was completely reorganised (mainly the order of the different paragraphs within the section) and renamed, to make our discussion about species-specific responses to 1) anoxia/sulphide and 2) food sources clearer.

3- In the same manner, in section 4.3 of the original submission (now 4.1), we moved the second paragraph (about previous studies investigating the foraminiferal response to the presence of sulphides) to a later part of the same section, and used it for comparison with our results.

4- The first paragraph of section 4.3 of the original submission (now 4.1), about previous publications investigating the response of foraminifera to low oxygen concentration/anoxia and presence of sulphide was moved to the introduction.

5- The third paragraph of section 4.4 of the original submission (now 4.2), about previous publications investigating the species-specific responses of foraminifera to low oxygen concentration/anoxia and presence of sulphide was also moved to the introduction.

6- The original section 4.1 (discussion) about Rose Bengal and CTG was shortened and incorporated in the introduction, as suggested by the two referees.

7- A new section 2.1 "Study area", which contains section 2.1 and 4.2 of the original submission presents the general environmental setting of Lake Grevelingen and the environmental parameters measured in the Den Osse Basin (where our studied stations are located).

8- In section 2.2 (about field sampling methodology) we added many methodological details as asked by both referees.

9- The first paragraph and a part of the second paragraph of section 3.4 of the original submission, about encrusted forms of E. magellanicum was moved, to a new section (2.6) in the Material and Methods part of the revised manuscript. The second part of the last paragraph of section 3.4 of the original submission about encrusted forms of *E. magellanicum* was enlarged and moved to discussion section 4.2.

**Referee #1**

Review of the manuscript bg-2019-382.

**General comments**

The manuscript entitled "Foraminiferal community response to seasonal anoxia in Lake Grevelingen (the Netherlands)" by Richirt et al. consists of a field study which aims to analyze the benthic foraminiferal community characteristics during 1.5 years, during when seasonal hypoxia/anoxia occurs together with presence of H2S. The results are very interesting and will be useful for the community, and the figures are clear and informative. However, the paper is poorly structured, and some statements need to be taken more carefully. Therefore, I suggest the following revisions before publication in Biogeosciences.

A restructuration of the material and methods section seems necessary. I suggest to create a studied area section, separated from the material and methods, with a description of the lake and the Den Osse Basin. It would be especially interesting as, as the authors say in the introduction, "a large amount of environmental data is available" (Line 64). This section should include all the references cited in the second part of section 2.1 (which should then be deleted) and in section 4.2 from the discussion. This section 4.2 is a description of already published results, not a discussion of the results from this paper. I suggest moving this paragraph to the studied area section.

Done, see general answer point 7.

The CTG method and a comparison with the Rose Bengal method was published in 2006, which is thirteen years ago. Since then, many studies have successfully used CTG to label their samples. I do not think that this part of the discussion about why the authors have chosen this method is necessary. The whole section 4.1 is a repetition of what you say in the introduction and does not bring anything new, it should be removed.

Done, see general answer point 6.

The discussion needs to be restructured as well. The first two paragraphs of section 4.3 are a description of the literature, not a discussion. I would move them in the introduction section, and cite these references where they are relevant, later in the discussion. Similarly in section 4.4, the 3rd paragraph is literature description that should or go in the introduction and/or be included later when discussing the results of the paper.

Done, see general answer points 4 and 5.

I strongly suggest to reconsider the whole structure of the section 4.4, which I find not easy to read in the current state.

Done, see general answer point 2.

The authors should be very careful about the reaction times they give in the discussion and conclusion. For example, line 329, the "two months after" cannot be assed for sure, as we do not have information about the fauna in October. Picking does take a lot of time, and I understand that picking all the months was not possible, but I recommend using more

approximate times, especially for station 1. Moreover, for me at station 1 on Fig. 11, the foraminiferal response to H2S appears immediate, as their abundance is lower already in July. Could the authors explain?

We agree with the reviewer, and changed the text accordingly (lines 305–309):

*"However, the fact that foraminiferal abundances reached almost zero only in September (about two months after the first occurrence of anoxic and sulphidic conditions in the upper sediment, in July) suggests that the presence of $H_2S$ did not cause instantaneous mortality, but that the disappearance of the foraminiferal community was a delayed response, probably caused by inhibited reproduction and, eventually, increased mortality."*

In the results, a full paragraph is dedicated to encrusted forms, together with a full plate of pictures, and two detailed graphs. I suggest to strongly develop this part in the discussion, which now consist of 4 lines, to include the information from the second paragraph of section 3.4 – and explanation given by these authors -, and the following references: Cedhagen 1996 (Phuket Marine Biological Center), and Heinz et al. 2005 (Marine Biology Research).

Done, see general answer point 9.
Because our results are merely an additional observation of this phenomenon, and do not allow us to draw any further conclusions about the function of these cysts, we do not want to develop this aspect much further in the discussion. However, we moved one sentence from the result section to the discussion section, and added some details about these cysts following the referee's suggestion.

Please also explain the current statements, do you suggest that the feeding cysts only get formed when *P. globosa* is blooming?

We found abundant encrusted forms (representing more than half of the specimens of *E. magellanicum*) only from May onwards, when the bloom of *P. globosa* occurs. This may of course be a coincidence, but it is also possible that the formation of feeding cysts (and then the proportion of encrusted individuals) is enhanced by the presence of suitable food, such as *P. globosa* may be for *E. magellanicum*.
In view of the lack of firm arguments in support of our rather speculative idea, we prefer not to develop it any further.

Please find below minor suggestions and text comments.
**Minor comments**

Abstract

Line 19: early diagenesis and organic matter recycling are mentioned here but never again in the paper. Please explain.

We think this is not really needed. However, we modified this sentence to underline the impact of these processes of the functioning of benthic ecosystems (lines 18–20):

*"These hypoxic events have large consequences for the functioning of benthic ecosystems. They profoundly modify early diagenetic processes involved in organic matter recycling, and in severe cases, they may lead to complete anoxia and presence of toxic sulphides in the*

*sediment and bottom water, thereby severely affecting biological compartments of benthic marine ecosystems."*

To:

*"Hypoxic events have large consequences for the functioning of benthic ecosystems. In severe cases, they may lead to complete anoxia and presence of toxic sulphides in the sediment and bottom-water, thereby strongly affecting biological compartments of benthic marine ecosystems."*

Line 30: This is in contradiction with your conclusion, is there not a "drop in standing stocks" for station 2?

We changed this sentence to clarify the relation between faunal density and euxinia (lines 28–30).

*"Conversely, at the shallower site (23 m), where the duration of anoxia and free $H_2S$ was shorter (one month or less), a dense foraminiferal community was found throughout the year."*

By:

*"Conversely, at the shallower site (23 m), where the duration of anoxia and free $H_2S$ was shorter (one month or less), a dense foraminiferal community was found throughout the year excepted for a short period after the stressful event."*

Line 34-35: The two sentences are in contradiction, please rephrase.

We do not see what is contradictory in our text.

Line 32: Replace "H2S" by "H2S".

Done

Introduction

Please shortly explain what are foraminifera, what are their place and role in these types of environment, and why you chose them for your study.

We do not think it is necessary to explain what are foraminifera; the aim of our study is explained in lines 82–83 and 97–98.

Line 43-46: This sentence is long and confusing, please rephrase.

We agree with the referee and rephrased it lines 42–45:

*"The combination of global warming and eutrophication is strengthening seasonal stratification of the water column, decreasing oxygen solubility, and enhancing benthic oxygen consumption in response to increased primary production, resulting from increased anthropogenic nutrient and/or organic matter input (i.e. eutrophication, Diaz and Rosenberg,*

*2008)."*

By:

*"This is due to the combination of (1) global warming, which is strengthening seasonal stratification of the water column and decreasing oxygen solubility and (2) eutrophication resulting from increased anthropogenic nutrient and/or organic matter input, which is enhancing benthic oxygen consumption in response to increased primary production (Diaz and Rosenberg, 2008)."*

Line 46: Could you give some examples of these consequences?

In view of the fact that the introduction is already very long, and that the biological response is widely known, we prefer to cite Riedel's review paper (line 46).

Line 50: I suggest to specify which ones of these references are field or culture studies, and to reconsider the sentence accordingly.

In order to answer together with the reviewer suggestion from lines 77–78, we replaced this list of references (which were moved lines 77–79) by a review from Koho et al., (2012) line 49:

*"Many foraminiferal taxa are able to withstand seasonal hypoxia/anoxia (e.g. Alve and Bernhard, 1995; Moodley et al., 1997; Moodley et al., 1998a; Geslin et al., 2004; Pucci et al., 2009; Koho et al., 2011; Langlet et al., 2013), and consequently can play a major role in carbon cycling in ecosystems affected by seasonal low oxygen concentrations (Woulds et al., 2007)."*

By:

*"Many foraminiferal taxa are able to withstand seasonal hypoxia/anoxia (see Koho et al., 2012 for a review), and consequently can play a major role in carbon cycling in ecosystems affected by seasonal low-oxygen concentrations (Woulds et al., 2007)."*

Line 52: Could you explain why/how anoxia and H2S are linked?

In Lake Grevelingen, the relation between anoxia and sulphide is very complex, because of the interference of cable bacteria. Explaining this here would take a lot of place, and would be somewhat superfluous. We added some references in which the relationship is treated in detail (Jørgensen 1982, Seitaj et al., 2015) line 51.

Lines 77-78: These references are already cited earlier. Please restructure.

All these references are now used once in the introduction in lines 77–79. See our answer to the reviewer suggestion line 50 just above.

We modified the sentence in order to specify if the study was from field or culture study lines 76–80:

*"Although the tolerance of foraminifera to low DO contents and long term anoxia (from weeks to 10 months) has been well documented for many species from different types of environments in laboratory culture (e.g. Moodley and Hess, 1992; Alve and Bernhard, 1995; Bernhard and Alve, 1996; Moodley et al., 1997; Duijnstee et al., 2003; Geslin et al., 2004; Duijnstee et al., 2005; Ernst et al., 2005; Pucci et al., 2009; Koho et al., 2011; Geslin et al., 2014) as well as in field studies (e.g. Piña-Ochoa et al., 2010b ; Langlet et al., 2013; 2014), their tolerance to free $H_2S$ is still debated."*

Line 81: Please add references, even if they are "sparse".

We removed this sentence, which concerns population dynamics. See general response, point 1.

Line 82: Is there not any previous foraminiferal studies in the lake itself?

There are only some previous reports (e.g. Donders et al. 2012), not published in peer-reviewed journals.

Lines 82-92: Please shorten this part, the CTG method is well known already.

In agreement with the second referee, we decided not to discuss Rose Bengal/CTG in the discussion section but rather to explain in the introduction why CTG is important particularly in environments where OM degradation may be very slow, lines 99–113.

Lines 96-97: This belongs to the method section, please remove.

This explains what we did so we think that it deserves to be mentioned at the end of the introduction. However, to take into account the reviewer's suggestion we changed the two sentences in lines 117–118:

*"Foraminiferal assemblages were studied in the top 1 cm layer. For each dominant species, size distributions were determined in order to get insight into the population dynamics."*

To:

*"Living foraminiferal assemblages were studied in the uppermost sediment and size distributions were determined in order to get insight into the possible moment(s) of reproduction or accelerated growth in test size."*

Line 100: example of these indices?

We think that adding this information would not be relevant.

Material and Methods

The description of the lake is not a part of the method. See also my general comments.

We agree with the reviewer. We added a "Studied area" section to the revised manuscript, see general answer point 7.

Please specify that SEM pictures were taken for the four dominant species including encrusted specimens, with which microscope and where were they taken.

We thank the reviewer for this comment. We added this in the Acknowledgements section in lines 500–502:

*"We are grateful to Romain Mallet and the team of the SCIAM imaging facility at the University of Angers."*
We also added T. Jauffrais and C. LeKieffre in acknowledgments.
*"We acknowledge Jassin Petersen for his help with recovering some of the environmental data **and Thierry Jauffrais and Charlotte LeKieffre for discussion about alternative metabolisms**."*

Lines 112-114: This paragraph should be moved to the field sampling section.

Done, see lines 148–150.

Line 114: I guess a map is available in the cited paper? Maybe you can precise it here?

We modified the sentence in agreement with the reviewer's suggestion line 149.
We also changed the reference here for an earlier one (Hagens et al., 2015):
*"Both station 1 (51°44.834' N, 3°53.401' E) and station 2 (51°44.956' N, 3°53.826' E) are located in the main channel, at 34 and 23 m depth, respectively (see map in Hagens et al, 2015)."*

Line 118: "similarly", by who?

We now precise it lines 155–156:

*"The data for station 2 (Supplementary Table 1) were acquired similarly and during the same cruises but never published, for further details about the sampling method, see Seitaj et al. (2015)."*

Line 120: Please give more details about the sampling. I see in the acknowledgments that the r/v Luctor was involved. What kind of corer was used? How long where the cores? Were some environmental data taken at the same time?

Done, see general answer point 8.

Line 123: I know that CTG labelling happens on the field. But after that you talk about picking. As this is not a field sampling event, I would move this to the sample treatment section.

Done, this sentence was moved to the section 2.3 (Sampled Treatment) on lines 171–173.

Line 127: Add "finally" before "investigated". See my comment about Table 1.

We changed the caption of Table 1 to make it clearer:

*"Sampling dates for stations 1 and 2. x = one core investigated, o = no core investigated"*

By:

*"Sampling dates of the samples which were investigated for living foraminifera for stations 1 and 2. x = one core investigated, o = no core investigated"*

Line 133: "previous studies", where were they? Please add references.

This is a generality. We took this value because it is easier to compare standardised volume with other studies. To make it clearer we changed line 176:
*"Abundances were then standardised to a volume of 10 cm$^3$ in order to facilitate comparison with previous studies."*
By:
*"Abundances were then standardised to a volume of 10 cm$^3$."*

Line 166: On which species was this done, and how many specimens were used?

Added in section 2.5 on lines 208–209:
*"The measurements were made for all species, which represent together 4176 individuals for station 1 and 19624 individuals for station 2."*

Results

Line 178-179: Remove this sentence. You already explain it in the method, and the Figure 2 has a caption.

Done.

Line 179: Please check if you mean total or mean abundances here.

We replaced *"Total abundances"* by *"Averaged total abundances"* line 241.

Line 183: I would be careful with the use of "early" and "late", talking about the seasons. July is not early summer. Line 433, March is not winter. Please check through the paper, maybe giving the months is the most accurate solution.

In the submitted version, we used astronomical seasons, in which March is late winter and July is early summer. As suggested by the reviewer, in order to avoid any confusion, we replaced and/or specified references to seasons in the manuscript with months in all necessary cases.

Lines 194-195: Please remove this sentence, you have already explained in the method.

Done.

Line 197: Replace "Fig.4" by "Figure 4". This sentence should be moved to the method section.

Done in line 257.

Lines 203-204: I suggest to remove this sentence, it does not bring anything new.

We don't really agree, even if the dominant species for both stations are given before, this sentence summarises the faunal difference between stations 1 and 2. We think it is a useful addition.

Line 206: Add "and Table 2" after Fig. 4.

Done in lines 265–266.

Line 211: Remove "(fairly low)"

*We removed the brackets. We changed this paragraph accordingly to the reviewer next comment.*

Line 213: We know that *T. inflata* was absent in 2011, as you said it line 205. Please rephrase. I think the way you described the results for the station 2 is clearer than for station 1. I suggest to also describe the station 1 species by species, instead of year by year.

The sentence in line 205 in the original submission is a general comparison between the two stations for all the species. Sentence line 213 in the original submission is dedicated to station 1 and only *Trochammina inflata*. We think that the two sentences have to remain in the manuscript.
We restructured the paragraph about station 1 and now describe densities species by species as suggested by the reviewer (lines 265–275).

Line 225: Please remove "Conversely to station 1", this is confusing here.

Done.

Lines 230-233: This should be moved to the methods section.

This part was moved to Supplementary material. See general answer point 1.

Line 234: Please add "(Fig. 6)" after "station 2".

Done.
This part was moved to Supplementary material. See general answer point 1.

Lines 256 and 258: same information, please modify.

Done.
This paragraph was modified following the general answer point 9.

Line 260: "Similar observations", where?

This paragraph was modified following the general answer point 9.
It is now in Discussion section lines 469–472.
To follow the reviewer comment we added the locations between the brackets:

*"Concerning the cysts of E. magellanicum described here, very similar observations have been made for Elphidium incertum at different locations (Norwegian Greenland Sea and*

*Baltic Sea in Linke and Lutze, 1993; Koljö Fjord in Gustafsson and Nordberg, 1999; Kiel Bight in Polovodova et al., 2009)."*

Line 260-262: This part should go to the discussion section.

*It is now in Discussion section lines 469–472.*
*See general answer point 9 and previous comment.*

Line 263: Please remove this sentence and cite the Figure 9 in the following sentence.

*To take into account the reviewer's suggestion, we modified the sentence in line 293–295:*
*"At station 1, encrusted forms of E. magellanicum were present in moderate proportions in May (26.8 % of the total E. magellanicum population, Fig. 9) and July (47.6 %); the species disappeared thereafter. At station 2, encrusted forms strongly dominated the E. magellanicum population from May (72.3 %) to December (88 %, Fig. 9)."*

Discussion

I think the section 4.1 should be removed from the paper. See also my general comments.

*See general discussion point 6.*

The information given in the 4.2 section are not results from this paper, they are a description of the site citing already published papers. This should go in the studied area section. See also my general comments.

*See general discussion point 7.*

In section 4.3, the actual discussion starts on line 321. See also my general comments.

*See general discussion point 3.*

Line 337: Do you have information about why the 2011 hypoxia was so severe compared to the 2012 one?

*We do not have information about this.*

Line 339: I know that this study only focus on living fauna, but it would have been interesting to check the dead fauna further down in the cores, to see if standing stocks were indeed higher before the 2011 severe hypoxia.

*This is a part of another paper about a long core studying dead fauna at station 1. Because the work is still in progress, we cannot say anything about this yet.*

Line 345: Could you explain how you deduced these "6 months" of recovery? As the hypoxia event was much more severe in 2011, how could we know if the H2S stayed longer in the upper sediment compared to 2012, and thus how long it affected the fauna? Please explain.

*We assumed that $H_2S$ front in the sediment migrated according to bottom water hypoxia in 2011 like in 2012 for stations 1 and 2: when bottom water hypoxia (or anoxia) occurred, $H_2S$ in the upper part of the sediment occurred also. See lines 337–340 in original submission and*

now lines 315–318.
We estimated a recovery time of about 6 months because this is the time between the resurgence of oxic conditions in the bottom water in September 2011 and presence of foraminifera in March 2012.

Line 366: Please remove "(i.e. like station 1)".

We removed this as suggested by the reviewer in line 355.

Line 365: This paragraph and the following one are very similar. I suggest to merge them.

In the first paragraph (lines 365-370 in original submission) we discuss the delayed response which also probably occurred in 2011.
In the second one (371-374 in original submission) we discuss the fact that repeated short events are probably more harmful than one short event (comparison 2011–2012).
Since these two ideas/suggestions are different, we prefer to keep the two paragraph as they are now in the discussion in lines 355–364.

Line 379: We cannot be sure about that, as there are no available data. Please modify this statement.

We want to point out that there are available data in November 2011 and January 2012 for station 1.
However, we slightly changed the sentence to make clearer that this is an assumption on lines 367–369:

*"However, at station 2, foraminiferal abundances increased again in December 2012, suggesting a recovery time of about two months, much shorter than at station 1, where standing stocks in the >125 μm fraction only increased 6 months after the presence of anoxia and free sulphides."*

By:

*"However, at station 2, foraminiferal abundances increased again in December 2012, suggesting a recovery time of about two months, **which is likely** much shorter than at station 1, where standing stocks in the >125 μm fraction only increased 6 months after the presence of anoxia and free sulphides."*

Line 386: Remove "(in contrast to station 2)".

We think it is a relevant comparison at this point in the discussion and we would like to keep this sentence as in original submission. Now in line 377.

Line 387: "by the nearby sites", I thought the water circulation was weak in the lake? Is transportation then possible? Please check.

This transport can happen because we are in one of the deepest channel of the lake, and could take place by under-water landslides.
Even in the case of weak water circulation, we cannot exclude completely transport of propagules from nearby sites for example.

We added *"possibly"* between the brackets on line 378:

*"(e.g. **possibly** by nearby sites or by the remaining few individuals)"*

Lines 395-396: This sentence belongs to the results section, please remove.

See general answer point 2.

Line 437: But no diatoms?

This information is not specified in Hagens et al. 2015.

Line 438: Which *Elphidium*? Elphidiids?

We refer to the genus *Elphidium* in general. For further information, see Pillet et al., 2011 as mentioned in the original submission.

Line 440-441: Remove this sentence, it is confusing there, and you talk about this aspect just after.

We think that this sentence has its place in the manuscript in its new form (see general discussion point 2) and that it should be conserved.

Line 470: What about *T. inflata*?

We decided not to discuss this species in this paragraph because we have no clue of its food source. To our knowledge, the general ecology is less known that that of the hyaline species of this study. Moreover, this species, although considered as dominant species based on subjective criteria (>1%), is less represented than the others. For these reasons we prefer not to discuss this species extensively in our manuscript.

Lines 476-479: This part should be developed. See also my general comments.

See general answer point 9. We developed this aspect in the discussion lines 466–475 as requested by the reviewer. However, we prefer not to discuss these encrusted forms in our manuscript further than this.

Conclusion

I would add a short introductory sentence or add details to the first sentence, to quickly remind the reader what you did.

We added a sentence in lines 473–474.
*"In this study we examined the foraminiferal community response to different durations of seasonal anoxia coupled with the presence of sulphide in the uppermost layer of sediment at two stations in Lake Grevelingen."*

References

Biogeosciences is very careful with bibliography details. Please go through your references list: ~10 papers miss doi, some miss page range, etc.

We carefully checked the references list as asked by the reviewer.

Figures
Table 1: You say in the text that the sampling happened every month, but that you only analyzed specific months. Thus, it the title correct here?

We corrected the caption, see previous referee comment about line 127.
We changed the caption of Table 1:
*"Sampling dates for stations 1 and 2. x = one core investigated, o = no core investigated"*
By:
*"Sampling dates of the samples which were investigated for living foraminifera for stations 1 and 2. x = one core investigated, o = no core investigated"*

Figure 1: In the caption, remove "This figure shows" in the first sentence, and add somewhere "for size measurement" as well as "ImageJ software".

We replaced:
*"This figure shows the different steps of the numerical treatment of each image. The left figure shows the untreated image, the middle figure presents the next step, when all individual foraminifera are depicted. Finally, the figure on the right shows the individual foraminiferal outlines which were measured."*
By:
*"Numerical treatment used for the size measurement for each image performed with ImageJ software. The left figure shows the untreated image, the middle figure presents the next step, when all individual foraminifera are depicted. Finally, the figure on the right shows the individual foraminiferal outlines which were measured."*

Figure 2: I don't think "Total living assemblage" is necessary below Station 1 and Station 2. Instead, it would be better to have this as the vertical axis title, with the unit (ind. 10 cm-3) into brackets. In the caption, replace "for which" by "where" to be consistent with other captions.

We replaced *"for which"* by *"where"* in the caption of figure 2.

However, we prefer not to remove *"Total living assemblage"* from the figure itself and place it as vertical axis title, because we think it helps the understanding and this is consistent with figures 11 and 12. We think that the figure is already clear enough when also considering caption.

Figure 4: Vertical axis title?

We think that the figure is already clear enough when also considering the caption.

Figure 5: Vertical axis title? Also, I guess you mean "station 1" in the last sentence. Please check the months in bold.

We think that the figure is already clear enough when also considering the caption.

We replaced *"station 2"* by *"station 1"* in the last sentence of the caption, as pointed out by the reviewer.

Figure 9: Vertical axis title? It would be informative to have the percentage of encrusted specimens on top of each bar.

We added the % of encrusted form specimens on top of each bar. We also changed the vertical axis legend *"ind. / 10 cm$^{-3}$"* by *"ind. 10 cm$^{-3}$"* to stay consistent with other figures. We think that the figure is already clear enough when also considering the caption concerning the vertical axis title.
The caption was modified accordingly:
*"Figure 9: Mean abundances (ind. 10 cm$^{-3}$) of non-encrusted (grey) and encrusted forms (black) of Elphidium magellanicum in 2012, at station, 1 (left) and 2 (right), with proportion of encrusted forms above each bar (in %). Investigated months are indicated in bold."*

Figure 11: You use the word "suboxic" in the caption, it's not coherent with the rest of the paper. Please check the months in bold.

We corrected the reference in the caption of figure 10 by replacing "Sulu-Gambari et al., 2015" by "Seitaj et al., 2015".
The term suboxic is used in Seitaj et al. 2015, the original publication where the data where published. We changed the caption as following:
*"The middle panel represents the depth (in mm) distribution of the oxic (blue), absence of oxygen and sulphides (orange) and sulphidic (black) zones within the sediment in 2012, from Seitaj et al. (2015)."*
The term "suboxic" does not appear anymore in the manuscript.

Figure 12: "Figure 12".

We changed *"Figure Discussion3"* by *"Figure 11"* (because figure 10 in the original submission was removed as asked by the second reviewer).

I hope my comments will be taken by the authors in a spirit of constructive criticism with only intention to further improve their manuscript.

Sincerely, Laurie M. Charrieau

**Referee #2**

Comments to the Author
Manuscript ID: bg-2019-382
The manuscript of Richirt and coauthors on "Foraminiferal community response to seasonal anoxia in Lake Grevelingen (the Netherlands)" represents the assemblage fluctuation of benthic community in response to hypoxic/anoxic environments. These analyses are important to understand the foraminiferal tolerance to hypoxia/anoxia and hydrogen sulfide, and to understand life histories under the extreme environments. However, the structure of manuscript is very poor and experimental design and data validations are problematic. Therefore, I strongly suggest reconstructing throughout the manuscript, and also data validation is needed.
The biggest problem is that authors only used specimens of 125μm or more. Juvenile specimens have the size smaller than 125μm. If you are looking at population dynamics, you must deal with juvenile specimens.

See general answer point 1.

We agree with the reviewer that the use of only the >125µm fraction (due to time limits) is a strong limitation and consequently, we cannot draw any firm conclusions about the population dynamics of the different species.

However, we already made very clear statements in the original submission itself that we use these data only to get insights and not to determine the population dynamics exhaustively.

In original submission section 2.5 line 165:

"In order **to gain insight** into the foraminiferal population dynamics, size measurements were performed on all samples of 2012"

We also already explained how we considered these results and emphasized the caution the reader must take when considering these data in the original submission section 2.5 lines 174-175:

"As we only examined the size fractions >125 µm, our analysis mainly concerns adult specimens, and does not include juveniles. **This limitation should be kept in mind when interpreting the results**."

Also in section 3.3 in the original version (now lines 223–230), we detailed what is possible or not possible to assess with our data and that we get only some clues concerning population dynamics in lines 245-251:

"Our tentative to distinguish cohorts by using a deconvolution method to separate the total size distributions into a sum of Gaussian curves was not conclusive. **The main problem was the fact that we did not have any information concerning individuals smaller than 125 µm**, so that our size distributions were systematically skewed on the left side (i.e. toward small individuals). An additional problem was the large number of smaller specimens which were always present. Because the identification of individual cohorts was not successful, **parameters like reproduction rate, growth rate or lifespan were not assessable**.

Nevertheless, **the size distribution data give some clues** concerning the population dynamics of the two dominant species.”

However, we agree that the term “population dynamics” for the foraminiferal size distribution in our study should not be used in the manuscript because it can be confusing, and we modified the manuscript accordingly.

In the methods section, the authors should explain more detailed procedures. I also found several methods sentences in the result and also in the discussion. Also, in the section 3.4, the authors described methods, although this paragraph is in the results section. These explanations should be move to appropriate section.

See general answer points 2, 8 and 9.

In the first section of the discussion (4.1), the author described both advantages and disadvantages of both CTG methods and Rose Bengal staining respectively. However, the CTG method has already described in Bernhard et al. (2006), and therefore it is not necessary to explain in detail.

See general answer points 6.

In the Section 4.2, I strongly suggest that the author should describe environmental setting of the sampling points. However, these descriptions must be explained in the beginning of this article. The authors also should explain vertical profile of oxygen in the water column and in the sediment in the "environmental settings of Den Osse Basin" section. This information can help readers to understand the habitat where foraminifera live in.

See general answer points 7.

In the section 4.3, I cannot understand what you want to discuss about. The authors referred (quoted) about previous studies in the first two paragraphs. The authors should move these paragraphs to the introduction, Ah...you would like to discuss relationship between sulfidic condition and foraminiferal assemblages? The discussion starts from line 321... I strongly recommend to make clear and re-structure throughout the manuscript.

See general answer points 3.

Other comments.

Line 34, “Elphidium selseyense and Elphidium magellanicum are much less affected by anoxia and free H2S than Ammonia sp. T6”
Is the light reaching the lake bottom? Is it not necessary to consider the photosynthesis of kleptoplasts?

We added the fact that light is not likely reaching the bottom of the lake in the discussion section 4.2 in lines 411–413:
*“As Hagens et al. (2015) observed that the light penetration depth in the Den Osse Basin never exceeded 15 m in 2012, and therefore photosynthesis by kleptoplasts (Bernhard and Alve, 1996) appears unlikely for both our aphotic stations (34 and 23 m depth).”*

Ammonia T6 has a nitrate pool in the cell. Nomaki et al. (2014, Limnol. Oceanogr., 59, 1879–1888) points out that this species potentially use an anaerobic respiration.

We now discuss about alternative metabolisms and symbiont bearing in introduction lines 93–96 and in discussion section 4.2 lines 395–403.

Line 47-, "Benthic faunas are strongly impacted by these events (Diaz and Rosenberg, 1995) although the meiofauna, especially foraminifera, appears to be less sensitive to low Dissolved Oxygen (DO) concentrations than the macrofauna"

Virgulinella, Bulimina, etc. may be sensitive to anoxic environments. Cannariato et al (1999, Geology, 27, 63-66) has analyzed community changes over the last 60,000 years at Santa Barbara Basin. As a result, low-oxygen torelant species are clearly replaced. Bolivina tumida, Buliminella tenuata and Globobulimina auriculata are low oxygen torelant species (dysoxic species). Interestingly, the response to hypoxia varies from species to species. Buliminella tenuata increase at the beginning of dysoxic. On the other hand, Bolivina tumida increases toward to the end of the dysoxic period.

We agree with the referee, this sentence is a general statement, and is completed by the next sentence which specifies that not all foraminiferal taxa are able to withstand with hypoxia/anoxia using also a reference (Koho et al., 2012) in lines 46–50:

*"Benthic faunas are strongly impacted by these events (Diaz and Rosenberg, 1995) although the meiofauna, especially foraminifera, appears to be less sensitive to low dissolved oxygen (DO) concentrations than the macrofauna (e.g. Josefson and Widbom, 1988).* **Many foraminiferal** *taxa are able to withstand seasonal hypoxia/anoxia (***see Koho et al., 2012 for a review***), and consequently can play a major role in carbon cycling in ecosystems affected by seasonal low-oxygen concentrations (Woulds et al., 2007)."*

Bolivina tumida has symbiotic microbes in its cells. Bolivina pacifica, Uvigerina peregrina, and Loxostomum pseudoberyichi retain microbes outside (in the pore) (Bernhard et al. 2018, Mar. Micropal. 138, 33-45). Based on these phenomena, it is expected that the response pattern to anoxia will differ depending on the symbiotic mode. The authors should explain/discuss this phenomenon in introduction and discussions.

We now discuss alternative metabolisms and symbiont bearing in the introduction in lines 93–96 and in the discussion in section 4.2 in lines 395–403.

Line53-, "Neutral molecular H2S can diffuse through cellular membranes and inhibits the functioning of cytochrome c oxydase (a mitochondrial enzyme involved in ATP production), finally inhibiting an aerobic respiration (Nicholls and Kim, 1982; Khan et al., 1990; Dorman et al., 2002)."
What do you think about an anaerobic respiration? The authors should explain about an anaerobic repiration.

We now discuss alternative metabolisms and symbiont bearing in the introduction in lines 93–96 and in the discussion in section 4.2 in lines 395–403.

Line 89, "In order to avoid this problem, we used CellTracker™ Green (CTG) to recognise living foraminifera. CTG is a fluorescent probe which marks only living individuals with

cytoplasmic (i.e. enzymatic) metabolic activity (Bernhard et al., 2006)"
This method is not new. The authors should only mention that CTG staining was used to distinguish live benthic foraminifera populations.

See general answer point 6.

Line 115-, "Measurements of oxygen concentrations in the bottom water (1 m above the sediment-water interface using a CTD) for 2011 are from Donders et al. (2012), whereas the 2012 data are from Hagens et al. (2015) and Seitaj et al (2017). Oxygen Penetration Depth (OPD) and depth of free H2 S detection were determined using O2 and H2S microsensors by Seitaj et al., (2015) for station 1, and the data for station 2 were acquired similarly (Supplementary Table 1)."
The authors should explain about environmental settings both station 1 and 2 in the beginning. This information can help reader to understand faunal assemblage changes (and population dynamics). This information is in the end of this manuscript.

See general answer point 7.

Line 121-, "The uppermost centimetre of each core was labelled with CellTracker™ Green CMFDA (CTG, 5-chloromethylfluorescein diacetate, final concentration of 1µM following Bernhard et al., 2006) and fixed in 5 % sodium borate buffered formalin after 24 h of incubation."
Where did you done this experiment? What kind of tools did you use? Did you sliced top 1cm and then put in the petri dish or some other container for CTG incubation? or jut put CTG directly onto the top of core? Need detailed experimental procedures.

See general answer point 8.

Line 129, "125µm"
As the authors mention about juvenile specimens, it is important point. Juvenile specimens have smaller than 125µm in size in many cases. If you are looking at population dynamics, you should deal with juvenile specimens. For this reason, it is difficult to see when the juvenile specimens have been reproduced.

See general answer point 1.

Line 145, "Supplementary Figure 1 shows…"
I found there are two types in these specimens. Specimens #145 and 152 have a larger proloculus than specimens #147 and #155. In my opinion, these differences in morphology correspond to different generations, megarospheric and microspheric. It is important points to find these generations to understand population dynamics. I strongly recommend to check which generations are abundant in each month.

Unfortunately, the scale bar in the previous version of the Supplementary Figure 1 was wrong for 3 of the 4 specimens. We corrected this, and measured a proloculus size of 43 to 61 µm, meaning that these are all megalospheric specimens. In fact, when we checked several of our microscopic slides, we found that the assemblages were always strongly dominated (>95%) by megalospheric specimens. In view of this, it doesn't seem useful to pay further attention to this point. All the more so, since we substantially diminished our discussion of population dynamics, with the few remaining elements now being presented as supplementary material.

Line 145, "the penultimate chamber"
Are there any differences in the pore size for each month (season)?

There is in fact a difference in pore size between stations 1 and 2, previously described by Petersen et al. (2016). We don't mention this, because we consider it to be outside the scope of the present paper.

Line 165, "population dynamics"
Need juvenile specimens for analyze.

Agree, see general answer point 1.

Line 169, "Fig. 1"
I think the authors should explain much more detail in this paragraph. Detailed procedures were written in figire1 caption!

Agree, see general answer point 8.

Line 175, ">125 μm, our analysis mainly concerns adult specimens, anddoes not include juveniles. This limitation should be kept in mind when interpreting the results"
If the authors discuss about population dynamics, it is necessary to check juveniles.

Agree, see general answer point 1.

Section3.1
Any statistical analyses?

We think that statistical analyses are not relevant/meaningless in view of the number of replicates and values (only 2).

Line 185, "very low in January"
I strongly suggest that the author should describe environmental setting of the sampling points. However, these descriptions must be explained in the beginning of this article. The authors also should explain vertical profile of oxygen in the water column and in the sediment in the "environmental settings of Den Osse Basin" section. This information can help readers to understand the habitat where foraminifera live in.

Agree, see general answer point 7.

Line 193, section 3.2
It is better to explain one by one. The authors should explain about station 1 and then explain about station 2.

This is already the case in original submission, lines 194–204: general statement
Lines 205–214: station 1
Lines 215–228: station 2

Line 221-, "then progressively decreased until the end of 2012 (= 48.1 ± 26) in November 2012). Trochammina inflata showed a similar pattern as Ammonia sp. T6"

It is necessary to indicate statistical analyses. Statistically significant?

In view of the low number of replicates, we think that inferential statistics will be not meaningful. We prefer to use descriptive statistics as the mean and standard deviation. However, we changed the word *"similar"* by *"analogous"* to soften this statement in line 283.

Line 237, "of larger individuals (>400 μm)"
Are there any ecological meanings?

Our discussion of the size distribution has been shortened and moved to supplementary material. In view of this, it doesn't seem useful to us to speculate about the ecological meaning of size differences.

Line 239-, "The low number of Ammonia sp. T6 individuals at station 1 does not allow us to draw any firm conclusion concerning the size distribution at this station"
In the result section, the authors should describe "results" in detail. For example, there are several large sized individuals in May, simultaneously 200~250 μm-sized individuals are there. How about propagules? Alve and Goldstein (2002, Journal of Micropaleontology, 21, 95-96; 2003, Limnology and Oceanography, 28, 2163-2170) discussed about propagules in their literatures.

See general comment point 1. This part was moved to the supplementary data. Therefore, it doesn't need more details.
Propagules are the same problematic than the fact that we do not look at specimens smaller than 125 µm. We also want to point out that looking at propagules in situ is very difficult because of taxonomical issues that arise when looking at very small individuals.

Line 243, "but started to diminish in December"
Are there any data? Please provide.

See general answer point 1.
Yes, see figure 7 right panel where the data are shown in original submission for station 2. Now moved in supplementary as Supplementary Figure 2.

Line 244, "decrease of the median to 339 μm"
Ammonia has two generations, asexual and sexual phases. These two generations are commonly found in spring and autumn. The authors have to think about the life cycle of foraminifera.

See our answer to the previous comment about micro/megalospheric alternance generation. There is no evidence of seasonal changes between megalospheric and microspheric generations in our material.

line 245-, "Our tentative to distinguish cohorts by using a deconvolution method to separate the total size distributions into a sum of Gaussian curves was not conclusive"
Please indicate in the methods section.

This paragraph was moved to the section 2.5 in lines 220–223:

*"In an attempt to recognize the different cohorts for each species in each of the bimonthly samples, we assumed that the size distribution was a sum of Gaussian curves, each of them representing a cohort. In order to identify the approximate mode for the Gaussian curves (i.e. cohorts), we used the changes in slope (i.e. inflexion points) of the second-order derivative of the total size distribution (Gammon et al., 2017)."*

Line 246-251,
It is not a result. If the goal is to evaluate foraminiferal behavior in an anaerobic environment, an experimental desing that analyzes small individuals should be considered. Objective 2 cannot be achieved.

We reformulated objective 2 in order to clarify that the aim of the paper is not to describe or explain population dynamics but rather the species-specific response to seasonal anoxia coupled with sulphide in lines 120–121:
 *"to obtain information about the life histories of the various species under adverse conditions"*
By:
*"to obtain information about the responses of the various species to adverse conditions."*

Line 255, "thin (Fig. 8c– e) and rather coarse"
Are there any data? To explain how it differs from the normal case, the authors should show the data.

Unfortunately, we don't have data concerning the thickness of the cysts. Since these cysts have only on very few occasions been described, and never in great detail, it is impossible to define what a "normal" cyst is.

Line 257-, "Because the crust stayed cohesive after exposition to 0.1 M of EDTA (EthyleneDiamineTetraacetic Acid) diluted in 0.1 M cacodylate buffer (acting as a carbonate chelator)"
This sentence should be moved to the methods section.

This sentence was moved to the Method section in lines 234–236:
*"In order to determine if the crust matrix is constituted of carbonate, we placed some specimens in microtubes and exposed them to 0.1 M of EDTA (EthyleneDiamineTetraacetic Acid) diluted in 0.1 M cacodylate buffer (acting as a carbonate chelator)."*

Line 259-,
This sentence should be move to the discussion section.

See also general answer point 9.
These sentences were modified and moved to the discussion section 4.2, lines 467–472.

Line 269-281,
It is not necessary to explain detailed about disadvantages of Rose Bengal staining method and advantages of CellTracker Green. Yes, the CellTracker Green labeling is suitable and reliable method to identify live specimens. However, incubation is required for the CTG method. I think this method includes some artifacts. During the staining, samples were transferred to petri dishes or bottles for 24 hours. The specimens were exposed different environmental condition from their habitat. This paragraph can be more shorten. Because this

method was already described in Bernhard et al (2006), so the authors do not need a detailed description of this method.

See general answer point 6.

Line 291, Fig. 10
You can omit this figure. Because, this is not your data. You can mark the timing of blooming on your figures 11 and 12.

We removed this figure from the manuscript and adapted the figure numbers in accordance with this change throughout the manuscript.

Line 328-,
In the case of symbiontic bacteria-bearing foraminifera, oxic condition is not suitable. Because symbiotic bacteria cannot consume hydrogen sulfide, methane or nitrate in an oxic condition, and the host foraminifera cannot use organic matter and/or anaerobic respiration from microbes.

We now discuss alternative metabolisms and symbiont bearing foraminifera in the introduction in lines 93–96 and in discussion section 4.2 lines 395–403.

Line 368-
There is little data in 2011. This sentence is overstatement. At both stations 1 and 2, low oxygen was observed from May to August. This situation is totally different from 2012. This characteristic situation will affect next year's (2012) assemblages.

In order to underline the speculative nature of our sentence, we modified lines 356-358 as follows:
*"If we assume that, like in 2012, rich foraminiferal faunas were present in spring 2011 at both stations, the low faunal densities observed in August and November 2011 could suggest that also in 2011, foraminifera show a delayed response to sulphidic conditions."*
To:
*"If we assume that, like in 2012, rich foraminiferal faunas were present in May–July 2011 at both stations, the low faunal densities observed in August and November 2011 could suggest that foraminifera **may have also shown** a delayed response to sulphidic conditions in 2011."*

We agree with the reviewer, that the succession of hypoxia was very different between the 2 years, but unfortunately, our faunal sampling in late 2011 is too scarce to compare the responses to the 2011 and 2012 hypoxia in detail.

Line 381-, "leading ultimately (in November) to almost complete disappearance of the foraminiferal fauna."
I'm worried about incubation time (duration) for CTG staining. For example, oxygen penetration depth is about 4mm in October at station 1, but sulfide layers still existed in the deep layer below 4mm. When the authors used top 1cm of the sediment for incubation, sulfidic conditions will be constructed in the experimental bottle (or other gear). For this reason, when living specimens still exist in top 4mm in October, sulfidic conditions may affect living ones. However, the authors did not explain detailed procedures of CTG staining methods. Long time exposure of sulfidic condition may affect living specimens. How did you evaluate for this effect in your experiment?

Immediately after sampling, and before adding the CTG stain, the sediment sample was carefully mixed with an equal volume of oxygenated water, and sample recipients were left unclosed in contact with the atmosphere. This treatment should be sufficient to oxidize all available sulphides.
The implicit question of the referee, whether in sulphidic conditions the metabolic activity of the foraminifera is still sufficient to be labelled with the CTG stain, is important. The answer may be different for different species, and can't be answered here.
We added details about sampling method as pointed out in the general answer point 8.

Line 384-, "inhibited reproduction, and eventually, increased mortality"
Need juvenile data.

See general answer point 1.
To take into account the reviewer's comment we changed this statement and slightly altered the sentence lines 373–375:
*"The delayed response at both stations shows that mortality has not been instantaneous, and suggests that the decreasing standing stocks are the result of inhibited reproduction, and eventually, increased mortality."*
By:
*"The delayed response at both stations shows that instantaneous mortality was limited, and suggests that the decreasing standing stocks might rather be the result of inhibited reproduction, and eventually, increased mortality."*

Line 390, Section 4.4
It is not appropriate section title. Need improvement. This section includes many topics related to environmental characteristics and food availability for foraminiferal responses. The authors should rearrange and clarify what authors want to discuss. This paragraph also includes the results. Need reconstruction.

We agree with the reviewer. We restructured this section as indicated in general answer point 2. We renamed the section as asked by the reviewer, which is now:
*"4.2 Species-specific response to anoxia, sulphide and food availability in Lake Grevelingen"*

Line 391-, 1st paragraph
Is this a topic sentence in this section? I think this information should be move to the Materials & Methods section.

See general answer point 2. The sentence was removed from the manuscript.

This section is also long and confusing. The authors have to reconstruct.

See general answer point 2.

Line 413, "take place throughout the year"
Are there any evidences that reproduction took place throughout the year? The authors should describe detailed results in the Result section. There exist relatively small-sized specimens that increased in May and September-October-November. In my opinion, it looks reproduction occurred twice in 2012. However, it is difficult conclude that there are no three or four chambered juveniles.

The suggestion of the reviewer is based on the increased number of Ammonia T6 specimens of 180 to 240 µm. However, these are already young adults. Unfortunately, we do not have any data for the 63-125 µm fraction, so that we can't draw firm conclusions about reproduction periods, as indicated by both referees.

---

## Author Comment (AC2) · 2 Dec 2019

[revised manuscript text omitted]

**Station 1**
*Elphidium selseyense* (n=3157)

**Station 2**
*Elphidium selseyense* (n=9583)

January

February

March

April

May

June

July

August

September

October

November

December

Maximum diameter (µm)

Number of individuals

**Figure 6:**  *Elphidium selseyense*

[Figure]

Figure 7: size distribution (maximum diameter for each individual in µm) of *Ammonia* sp. T6 for stations 1 (left) and 2 (right) in 2012. For each month, the number of individuals (n), the mode and the number of individuals associated to the mode (between brackets) are indicated in black. The medians are indicated by the red bars in each panel.

[Figure]

1065 **Figure 8: SEM images of (a) fully encrusted specimen, (b) partially encrusted specimen, (c) crushed encrusted specimen of *Elphidium magellanicum*. Note the thinness of the crust and the spinose structures on (d) and (e).**

[Figure]

**Figure 9:** Mean abundances (ind. 10 cm⁻³) of non-encrusted (grey) and encrusted forms (black) of *Elphidium magellanicum* in 2012, at station, 1 (left) and 2 (right), with proportion of encrusted forms above each bar (in %). Investigated months are indicated in bold.

[Figure]

[Figure]

**Figure 10:**

[Figure]

**Figure 11:** The top panel represents bottom‑-water oxygen concentrations (μmol L[-1]) in 2011 and 2012 at station 1, from Donders et al. (2012) and Seitaj et al. (2017). The grey horizontal dotted line indicates the hypoxia limit (63 μmol L[-1]). The middle panel represents the depth (in mm) distribution of the oxic (blue), absence of oxygen and sulphides (orange,) and sulphidic (black) zones within the sediment in 2012, from Seitaj et al. (2015). The bottom panel shows the total living foraminiferal abundances for both replicates (grey bars), mean abundances (diamonds) and standard deviations (black error bars) calculated for the two replicates, for all investigated months (in bold) in 2011 and 2012.

[Figure]

**Figure 1211:** The top panel represents bottom-water oxygen concentrations ($\mu$mol $L^{-1}$) in 2011 and 2012 at station 2, from Donders et al. (2012) and Seitaj et al. (2017). The grey horizontal dotted line indicates the hypoxia limit (63 $\mu$mol $L^{-1}$). The middle panel represents the depth (in mm) distribution of the oxic (blue), suboxic (orange, absence of oxygen and sulphides) and sulphidic (black) zones within the sediment in 2012. The bottom panel shows the total living foraminiferal abundances for both replicates (grey bars), mean abundances (diamonds) and standard deviations (black error bars) calculated for the two replicates, for all investigated months (in bold) in 2011 and 2012.

[Figure]

*Associated with the manuscript:*

**Foraminiferal community response to seasonal anoxia in Lake Grevelingen (the Netherlands)**

Julien Richirt[1], Bettina Riedel[1,2], Aurélia Mouret[1], Magali Schweizer[1], Dewi Langlet[1,3], Dorina Seitaj[4], Filip J. R. Meysman[5,6], Caroline P. Slomp[7] and Frans J. Jorissen[1]

[1]UMR 6112 LPG-BIAF Recent and Fossil Bio-Indicators, University of Angers, 2 Boulevard Lavoisier, F-49045 Angers, France
[2]First Zoological Department, Vienna Museum of Natural History, Burgring 7, 1010 Vienna, Austria
[3]Univ. Lille, CNRS, Univ. Littoral Côte d'Opale, UMR 8187, LOG, Laboratoire d'Océanologie et de Géosciences, F 62930 Wimereux, France
[4]Department of Ecosystem Studies, Royal Netherlands Institute for Sea Research (NIOZ), Yerseke, the Netherlands
[5]Department of Biology, University of Antwerp, Universiteitsplein 1, BE-2610 Wilrijk, Belgium
[6]Department of Biotechnology, Delft University of Technology, 2629 HZ Delft, the Netherlands
[7]Department of Earth Sciences (Geochemistry), Faculty of Geosciences, Utrecht University, Princetonlaan 8a, 3584 CB Utrecht, the Netherlands

*Correspondence to*: Julien Richirt (richirt.julien@gmail.com)

Supplementary figure 1. SEM images of spiral side and a 1000x magnification of the penultimate chamber for four individuals from Grevelingen station 1 identified T6 by molecular identification.

**Supplementary Table 1. Oxygen Penetration Depth ± sd and free H2S detection depth ± sd for each month in 2012 for both stations 1 and 2 (in mm).**

| Station | Month | OPD (mm) | H$_2$S depth (mm) |
|---------|-------|----------|-------------------|
| Station 1 | January | 1.7 ± 0.3 | 16.5 ± 3.2 |
| | February | 2 ± 0.4 | 17.1 ± 2.8 |
| | March | 1.7 ± 0.3 | 17.5 ± 0.7 |
| | April | 1 ± 0.2 | 18.6 ± 4.8 |
| | May | 1 ± 0.1 | 9.9 ± 2.2 |
| | June | 0.9 ± 0.1 | 7.9 ± 5.3 |
| | July | 0 ± 0 | 0.1 ± 0.1 |
| | August | 0 ± 0 | 0.9 ± 1.1 |
| | September | 0.7 ± 0.1 | 0.3 ± 0.2 |
| | October | 1.1 ± 0.1 | 3.3 ± 1.1 |
| | November | 0.4 ± 0 | 10.3 ± 1.9 |
| | December | 1.1 ± 0.2 | 13.4 ± 1.8 |
| Station 2 | January | 2.8 ± 0 | 19.6 ± 2 |
| | February | 2.4 ± 0.2 | 15.8 ± 1.2 |
| | March | 2.6 ± 0.6 | 20.3 ± 3.3 |
| | April | 1.4 ± 0.2 | 23.3 ± 0.3 |
| | May | 1.6 ± 0 | 26.4 ± 1 |
| | June | 1.1 ± 0.4 | 17.1 ± 0.4 |
| | July | 1.3 ± 0.4 | 1.1 ± 0.8 |
| | August | 0 ± 0 | 0.4 ± 0.2 |
| | September | 1.2 ± 0.2 | 0.8 ± 0.2 |

| | | |
|---|---|---|
| October | 1.6 ± 0.3 | 6.4 ± 2.9 |
| November | 1.3 ± 0.2 | 9.1 ± 3.3 |
| December | 1.5 ± 0.2 | 9.2 ± 0.7 |

**Supplementary Table 2. Living foraminiferal abundances for each replicate for the dominant species and total assemblage (ind./10cm3).**

**STATION 1**

| Species | | *Elphidium selseyense* | | *Ammonia* sp. *T6* | | *Elphidium magellanicum* | | *Trochammina inflata* | | Total assemblage | |
|---|---|---|---|---|---|---|---|---|---|---|---|
| Year | Month | A | B | A | B | A | B | A | B | A | B |
| 2011 | August | 2.1 | 0.4 | 1.4 | 1.1 | 0.0 | 0.0 | 0.0 | 0.0 | 4.2 | 2.5 |
| 2011 | November | 0.0 | 1.1 | 0.0 | 0.7 | 0.0 | 0.0 | 0.0 | 0.0 | 0.0 | 2.1 |
| 2012 | January | 2.8 | 7.4 | 0.7 | 5.7 | 0.0 | 0.4 | 0.4 | 2.1 | 5.0 | 18.0 |
| 2012 | March | 28.6 | 19.1 | 12.0 | 13.8 | 29.4 | 13.8 | 2.1 | 0.7 | 75.7 | 48.5 |
| 2012 | May | 141.5 | 531.6 | 13.8 | 4.6 | 63.0 | 129.8 | 0.4 | 3.2 | 222.1 | 677.6 |
| 2012 | July | 76.0 | 247.9 | 8.1 | 12.4 | 3.9 | 3.5 | 0.0 | 0.0 | 88.4 | 270.6 |
| 2012 | September | 21.2 | 38.2 | 0.7 | 3.9 | 0.0 | 0.0 | 0.0 | 0.7 | 21.9 | 46.0 |
| 2012 | November | 0.7 | 1.4 | 0.4 | 0.4 | 0.0 | 0.0 | 0.0 | 0.0 | 1.4 | 1.8 |

**STATION 2**

| Species | | *Elphidium selseyense* | | *Ammonia* sp. *T6* | | *Elphidium magellanicum* | | *Trochammina inflata* | | Total assemblage | |
|---|---|---|---|---|---|---|---|---|---|---|---|
| Year | Month | A | B | A | B | A | B | A | B | A | B |
| 2011 | August | 53.8 | 95.8 | 72.5 | 91.6 | 0.0 | 0.0 | 10.6 | 18.7 | 140.1 | 208.0 |
| 2011 | November | 33.2 | 71.4 | 61.9 | 59.8 | 0.0 | 0.0 | 13.1 | 10.6 | 111.1 | 146.4 |
| 2012 | January | 122.0 | 201.6 | 263.1 | 189.2 | 1.1 | 0.7 | 142.5 | 100.4 | 545.4 | 501.9 |
| 2012 | March | 225.6 | 203.7 | 275.2 | 152.8 | 41.0 | 56.6 | 73.9 | 76.0 | 624.2 | 500.5 |
| 2012 | May | 254.6 | 321.8 | 165.9 | 128.4 | 120.6 | 111.4 | 42.1 | 30.1 | 602.3 | 607.3 |
| 2012 | July | 318.3 | 246.9 | 172.2 | 144.7 | 39.6 | 36.1 | 35.4 | 27.6 | 589.9 | 473.2 |
| 2012 | September | 415.2 | 315.8 | 141.1 | 63.7 | 97.3 | 46.7 | 14.9 | 17.3 | 681.2 | 453.8 |
| 2012 | October | 104.7 | 92.7 | 87.0 | 111.1 | 2.1 | 1.4 | 5.3 | 9.5 | 205.8 | 217.2 |

| | | | | | | | | | | |
|---|---|---|---|---|---|---|---|---|---|---|
| 2012 | November | 29.4 | 32.5 | 66.5 | 29.7 | 3.9 | 4.2 | 5.0 | 2.5 | 108.9 | 73.2 |
| 2012 | December | 281.2 | 223.2 | 78.9 | 77.1 | 16.3 | 34.7 | 15.9 | 9.5 | 405.3 | 350.5 |

**Supplementary Table 3. Living foraminiferal abundances for each replicate, year and month for all the species of the assemblage (ind./10cm3).**

| Year | Station | Replicate | Month | Ammonia falsobeccarii (T15) | Ammonia sp. T1 | Ammonia sp. T2 | Ammonia sp. T3 | Ammonia sp. T6 | Bulimina denudata | Bulimina elongata | Bulimina marginata | Bulimina sp. | Cassidulina sp. | Elphidium selseyense | Elphidium magellanicum | Elphidium magellanicum (encrusted) | Elphidium margaritaceum | Elphidium sp. | Epistominella sp. | Haynesina depressula | Haynesina germanica | Hopkinsina sp. | Leptohalysis sp. | Non determined | Nonion sp. | Nonionella sp. | Quinqueloculina leavigata | Quinqueloculina sp. | Stainforthia sp. | Textularia sp. | Trochammina inflata |
|---|---|---|---|---|---|---|---|---|---|---|---|---|---|---|---|---|---|---|---|---|---|---|---|---|---|---|---|---|---|---|---|
| 2011 | 1 | A | August | 0.0 | 0.0 | 0.0 | 0.0 | 1.4 | 0.0 | 0.0 | 0.0 | 0.0 | 0.0 | 2.1 | 0.0 | 0.0 | 0.0 | 0.0 | 0.0 | 0.0 | 0.0 | 0.0 | 0.0 | 0.4 | 0.0 | 0.0 | 0.0 | 0.4 | 0.0 | 0.0 | 0.0 |
| 2011 | 1 | A | November | 0.0 | 0.0 | 0.0 | 0.0 | 0.0 | 0.0 | 0.0 | 0.0 | 0.0 | 0.0 | 0.0 | 0.0 | 0.0 | 0.0 | 0.0 | 0.0 | 0.0 | 0.0 | 0.0 | 0.0 | 0.0 | 0.0 | 0.0 | 0.0 | 0.0 | 0.0 | 0.0 | 0.0 |
| 2012 | 1 | A | January | 0.0 | 0.0 | 0.0 | 0.0 | 0.7 | 0.0 | 0.0 | 0.0 | 0.0 | 0.0 | 2.8 | 0.0 | 0.0 | 0.0 | 0.0 | 0.0 | 1.1 | 0.0 | 0.0 | 0.0 | 0.0 | 0.0 | 0.0 | 0.0 | 0.0 | 0.0 | 0.0 | 0.4 |
| 2012 | 1 | A | March | 0.4 | 0.0 | 1.1 | 0.0 | 12.0 | 0.4 | 0.0 | 0.0 | 0.0 | 0.0 | 28.6 | 29.4 | 0.0 | 0.0 | 0.4 | 0.0 | 0.4 | 0.0 | 0.0 | 0.0 | 0.0 | 0.0 | 0.0 | 0.0 | 0.7 | 0.0 | 0.4 | 2.1 |
| 2012 | 1 | A | May | 0.0 | 0.0 | 0.0 | 0.0 | 13.8 | 1.1 | 0.0 | 0.4 | 0.0 | 0.0 | 141.5 | 47.7 | 15.2 | 0.0 | 0.0 | 0.0 | 0.0 | 0.0 | 0.0 | 0.0 | 0.4 | 0.0 | 0.4 | 1.1 | 0.0 | 0.4 | 0.0 | 0.4 |
| 2012 | 1 | A | July | 0.0 | 0.0 | 0.0 | 0.0 | 8.1 | 0.0 | 0.0 | 0.0 | 0.0 | 0.0 | 76.0 | 1.8 | 2.1 | 0.0 | 0.0 | 0.0 | 0.0 | 0.0 | 0.0 | 0.0 | 0.0 | 0.0 | 0.0 | 0.0 | 0.0 | 0.0 | 0.4 | 0.0 |
| 2012 | 1 | A | September | 0.0 | 0.0 | 0.0 | 0.0 | 0.7 | 0.0 | 0.0 | 0.0 | 0.0 | 0.0 | 21.2 | 0.0 | 0.0 | 0.0 | 0.0 | 0.0 | 0.0 | 0.0 | 0.0 | 0.0 | 0.0 | 0.0 | 0.0 | 0.0 | 0.0 | 0.0 | 0.0 | 0.0 |
| 2012 | 1 | A | November | 0.0 | 0.0 | 0.4 | 0.0 | 0.4 | 0.0 | 0.0 | 0.0 | 0.0 | 0.0 | 0.7 | 0.0 | 0.0 | 0.0 | 0.0 | 0.0 | 0.0 | 0.0 | 0.0 | 0.0 | 0.0 | 0.0 | 0.0 | 0.0 | 0.0 | 0.0 | 0.0 | 0.0 |
| 2011 | 1 | B | August | 0.0 | 0.0 | 0.0 | 0.0 | 1.1 | 0.0 | 0.0 | 0.0 | 0.0 | 0.0 | 0.4 | 0.0 | 0.0 | 1.1 | 0.0 | 0.0 | 0.0 | 0.0 | 0.0 | 0.0 | 0.0 | 0.0 | 0.0 | 0.0 | 0.0 | 0.0 | 0.0 | 0.0 |
| 2011 | 1 | B | November | 0.0 | 0.0 | 0.0 | 0.0 | 0.7 | 0.0 | 0.0 | 0.0 | 0.0 | 0.0 | 1.1 | 0.0 | 0.0 | 0.0 | 0.0 | 0.0 | 0.0 | 0.0 | 0.0 | 0.0 | 0.0 | 0.0 | 0.0 | 0.4 | 0.0 | 0.0 | 0.0 | 0.0 |
| 2012 | 1 | B | January | 0.0 | 0.0 | 0.7 | 0.0 | 5.7 | 0.0 | 0.0 | 0.0 | 0.0 | 0.0 | 7.4 | 0.4 | 0.0 | 0.4 | 0.0 | 0.0 | 0.0 | 0.0 | 0.0 | 0.0 | 0.0 | 0.0 | 0.0 | 1.1 | 0.4 | 0.0 | 0.0 | 2.1 |
| 2012 | 1 | B | March | 0.0 | 0.0 | 0.0 | 0.0 | 13.8 | 0.0 | 0.0 | 0.0 | 0.0 | 0.0 | 19.1 | 13.8 | 0.0 | 0.0 | 0.0 | 0.0 | 0.4 | 0.0 | 0.0 | 0.0 | 0.4 | 0.0 | 0.0 | 0.4 | 0.0 | 0.0 | 0.0 | 0.7 |
| 2012 | 1 | B | May | 0.0 | 0.0 | 0.0 | 0.0 | 4.6 | 0.4 | 0.0 | 0.0 | 0.0 | 0.0 | 531.6 | 93.4 | 36.4 | 0.4 | 0.0 | 0.7 | 0.4 | 0.0 | 0.0 | 0.0 | 2.1 | 0.0 | 0.4 | 0.4 | 1.1 | 2.5 | 0.4 | 3.2 |
| 2012 | 1 | B | July | 0.0 | 0.0 | 0.4 | 0.0 | 12.4 | 0.4 | 0.0 | 0.7 | 0.0 | 0.0 | 247.9 | 2.1 | 1.4 | 1.4 | 0.4 | 0.0 | 0.0 | 0.0 | 0.0 | 0.0 | 0.7 | 0.0 | 0.0 | 0.7 | 0.4 | 1.8 | 0.0 | 0.0 |
| 2012 | 1 | B | September | 0.0 | 0.0 | 0.0 | 0.0 | 3.9 | 0.0 | 0.0 | 0.0 | 0.0 | 0.0 | 38.2 | 0.0 | 0.0 | 0.4 | 0.0 | 0.0 | 0.0 | 0.0 | 0.0 | 0.0 | 0.0 | 0.0 | 0.0 | 0.0 | 0.0 | 2.5 | 0.4 | 0.7 |
| 2012 | 1 | B | November | 0.0 | 0.0 | 0.0 | 0.0 | 0.4 | 0.0 | 0.0 | 0.0 | 0.0 | 0.0 | 1.4 | 0.0 | 0.0 | 0.0 | 0.0 | 0.0 | 0.0 | 0.0 | 0.0 | 0.0 | 0.0 | 0.0 | 0.0 | 0.0 | 0.0 | 0.0 | 0.0 | 0.0 |
| 2011 | 2 | A | August | 0.0 | 0.0 | 0.0 | 0.0 | 72.5 | 0.0 | 0.0 | 0.0 | 0.0 | 0.0 | 53.8 | 0.0 | 0.0 | 0.7 | 0.0 | 0.0 | 0.0 | 0.0 | 0.4 | 0.0 | 1.1 | 0.0 | 0.0 | 0.4 | 0.0 | 0.4 | 0.0 | 10.6 |
| 2011 | 2 | A | November | 0.0 | 0.0 | 0.0 | 0.0 | 61.9 | 0.0 | 0.0 | 0.0 | 0.0 | 0.0 | 33.2 | 0.0 | 0.0 | 0.7 | 0.0 | 0.0 | 0.0 | 0.0 | 0.0 | 0.0 | 1.1 | 0.0 | 0.0 | 1.1 | 0.0 | 0.0 | 0.0 | 13.1 |
| 2012 | 2 | A | January | 0.7 | 0.0 | 2.5 | 8.8 | 263.1 | 0.0 | 1.1 | 0.0 | 0.0 | 0.0 | 122.0 | 1.1 | 0.0 | 0.7 | 0.4 | 1.1 | 0.0 | 0.0 | 0.0 | 0.0 | 0.0 | 0.0 | 0.7 | 0.4 | 0.0 | 0.0 | 0.4 | 142.5 |
| 2012 | 2 | A | March | 0.0 | 0.0 | 1.4 | 0.0 | 275.2 | 0.0 | 0.0 | 0.0 | 1.8 | 0.0 | 225.6 | 40.0 | 1.1 | 0.4 | 0.0 | 0.4 | 0.0 | 0.0 | 0.0 | 0.0 | 0.0 | 0.0 | 0.7 | 0.7 | 0.0 | 1.4 | 1.8 | 73.9 |
| 2012 | 2 | A | May | 0.0 | 0.0 | 1.1 | 0.0 | 165.9 | 0.0 | 0.0 | 0.4 | 3.9 | 0.0 | 254.6 | 38.6 | 82.1 | 0.4 | 0.0 | 1.4 | 0.0 | 0.0 | 0.0 | 0.0 | 0.0 | 0.0 | 0.0 | 3.2 | 0.4 | 2.1 | 1.4 | 42.1 |
| 2012 | 2 | A | July | 0.0 | 0.0 | 1.8 | 0.0 | 172.2 | 6.0 | 2.1 | 0.4 | 0.4 | 0.0 | 318.3 | 3.9 | 35.7 | 1.4 | 0.0 | 0.4 | 0.7 | 0.0 | 0.0 | 0.0 | 0.0 | 0.0 | 0.0 | 0.4 | 0.0 | 7.1 | 1.8 | 35.4 |

| Year | | | Month | V1 | V2 | V3 | V4 | V5 | V6 | V7 | V8 | V9 | V10 | V11 | V12 | V13 | V14 | V15 | V16 | V17 | V18 | V19 | V20 | V21 | V22 | V23 | V24 | V25 | V26 | V27 | V28 |
|---|---|---|---|---|---|---|---|---|---|---|---|---|---|---|---|---|---|---|---|---|---|---|---|---|---|---|---|---|---|---|---|
| 2012 | 2 | A | September | 0.0 | 0.7 | 0.0 | 0.0 | 141.1 | 0.0 | 1.4 | 0.4 | 0.0 | 0.0 | 415.2 | 16.3 | 81.0 | 0.4 | 0.4 | 3.2 | 0.0 | 0.0 | 1.4 | 0.0 | 0.0 | 0.0 | 0.0 | 0.0 | 0.4 | 1.4 | 3.2 | 14.9 |
| 2012 | 2 | A | October | 0.0 | 0.4 | 0.7 | 0.0 | 87.0 | 1.1 | 2.5 | 0.4 | 0.0 | 0.0 | 104.7 | 0.0 | 2.1 | 0.0 | 0.0 | 0.0 | 0.0 | 0.0 | 0.0 | 0.4 | 0.0 | 0.0 | 0.0 | 0.0 | 0.0 | 0.0 | 1.4 | 5.3 |
| 2012 | 2 | A | November | 0.0 | 0.0 | 0.0 | 0.0 | 66.5 | 0.7 | 0.0 | 0.4 | 0.0 | 0.0 | 29.4 | 0.0 | 3.9 | 0.4 | 0.0 | 0.0 | 0.0 | 0.0 | 2.1 | 0.0 | 0.0 | 0.0 | 0.0 | 0.0 | 0.0 | 0.7 | 0.0 | 5.0 |
| 2012 | 2 | A | December | 0.7 | 0.0 | 1.8 | 0.0 | 78.9 | 1.1 | 0.7 | 1.4 | 0.0 | 0.0 | 281.2 | 0.4 | 15.9 | 0.0 | 0.0 | 0.7 | 0.4 | 0.0 | 1.8 | 0.0 | 0.4 | 0.0 | 0.4 | 0.0 | 0.4 | 0.4 | 3.2 | 15.9 |
| 2011 | 2 | B | August | 0.0 | 0.0 | 0.0 | 0.0 | 91.6 | 0.0 | 0.0 | 0.0 | 0.4 | 0.0 | 95.8 | 0.0 | 0.0 | 0.0 | 0.0 | 0.0 | 0.7 | 0.4 | 0.0 | 0.0 | 0.4 | 0.0 | 0.0 | 0.0 | 0.0 | 0.0 | 0.0 | 18.7 |
| 2011 | 2 | B | November | 0.0 | 0.0 | 0.0 | 0.0 | 59.8 | 0.0 | 0.0 | 0.0 | 0.4 | 0.0 | 71.4 | 0.0 | 0.0 | 1.1 | 0.0 | 0.0 | 1.1 | 0.0 | 0.0 | 0.0 | 1.1 | 0.0 | 0.0 | 0.0 | 1.1 | 0.0 | 0.0 | 10.6 |
| 2012 | 2 | B | January | 0.0 | 0.4 | 2.1 | 0.0 | 189.2 | 0.0 | 0.4 | 0.0 | 0.0 | 0.0 | 201.6 | 0.7 | 0.0 | 0.0 | 1.1 | 0.0 | 0.0 | 0.0 | 0.0 | 0.0 | 0.0 | 0.0 | 0.0 | 0.0 | 5.7 | 0.0 | 0.4 | 100.4 |
| 2012 | 2 | B | March | 0.0 | 0.0 | 1.1 | 0.0 | 152.8 | 0.4 | 0.0 | 0.0 | 2.1 | 0.0 | 203.7 | 56.2 | 0.4 | 1.1 | 0.7 | 1.4 | 0.0 | 0.0 | 0.0 | 0.0 | 0.0 | 1.1 | 0.4 | 0.0 | 1.8 | 0.7 | 0.7 | 76.0 |
| 2012 | 2 | B | May | 0.0 | 0.0 | 1.4 | 0.0 | 128.4 | 2.1 | 0.0 | 0.7 | 0.0 | 0.4 | 321.8 | 25.8 | 85.6 | 0.0 | 0.0 | 0.4 | 0.4 | 0.0 | 0.0 | 0.0 | 1.8 | 0.0 | 2.8 | 1.1 | 0.7 | 1.1 | 2.8 | 30.1 |
| 2012 | 2 | B | July | 0.0 | 1.1 | 1.4 | 0.0 | 144.7 | 0.4 | 1.8 | 1.8 | 2.1 | 0.0 | 246.9 | 8.1 | 27.9 | 0.7 | 0.0 | 1.1 | 1.1 | 0.0 | 0.0 | 0.0 | 0.0 | 0.4 | 2.1 | 1.1 | 0.7 | 2.5 | | 27.6 |
| 2012 | 2 | B | September | 0.0 | 0.0 | 0.4 | 0.0 | 63.7 | 1.8 | 0.7 | 0.0 | 0.0 | 0.0 | 315.8 | 8.1 | 38.6 | 1.4 | 0.4 | 2.1 | 0.0 | 0.4 | 0.0 | 0.0 | 0.0 | 0.0 | 0.0 | 0.0 | 0.4 | 1.4 | 1.4 | 17.3 |
| 2012 | 2 | B | October | 0.0 | 0.7 | 1.1 | 0.0 | 111.1 | 0.4 | 0.0 | 0.0 | 0.0 | 0.0 | 92.7 | 1.1 | 0.4 | 0.0 | 0.0 | 0.4 | 0.0 | 0.0 | 0.0 | 0.0 | 0.0 | 0.0 | 0.0 | 0.0 | 0.0 | 0.0 | 0.0 | 9.5 |
| 2012 | 2 | B | November | 0.0 | 0.0 | 0.4 | 0.0 | 29.7 | 1.1 | 0.0 | 0.4 | 0.0 | 0.0 | 32.5 | 1.8 | 2.5 | 0.4 | 0.0 | 0.7 | 0.0 | 0.0 | 0.0 | 0.0 | 0.0 | 0.0 | 0.0 | 0.4 | 0.4 | 0.7 | | 2.5 |
| 2012 | 2 | B | December | 0.0 | 0.0 | 0.0 | 0.0 | 77.1 | 1.4 | 0.7 | 0.0 | 0.0 | 0.0 | 223.2 | 5.7 | 29.0 | 1.1 | 0.0 | 1.4 | 0.0 | 0.0 | 0.4 | 0.0 | 0.0 | 0.0 | 0.0 | 0.4 | 0.4 | 0.0 | 0.4 | 9.5 |

[Figure]

Supplementary figure 1. SEM images of spiral side and a 1000x magnification of the penultimate chamber for four individuals from Grevelingen station 1 identified T6 by molecular identification.

[Figure]

**Supplementary Figure 2: A:** size distribution (maximum diameter for each individual in μm) of *Elphidium selseyense* for stations 1 (left) and 2 (right) in 2012. **B:** size distribution (maximum diameter for each individual in μm) of *Ammonia* sp. T6 for stations 1 (left) and 2 (right) in 2012. For each month, the number of individuals (n), the mode and the number of individuals associated to the mode (between brackets) are indicated in black. The medians are indicated by the red bars in each panel. In order to base our analysis on a sufficiently high number of specimens, we focused on *E. selseyense* and *Ammonia* sp. T6. As explained before, we only considered specimens retained on a 125 μm mesh meaning that juvenile specimens are not represented. Only the samples taken in 2012 were considered. The size distribution of *E. selseyense* was relatively similar between the two stations regarding the median, ranging from 253 μm (in May) to 295 μm (in November) at station 1 and from 261 μm (in October) to 290 μm (in March) at station 2. At both stations, we observed the presence of an abundant group of smaller specimens, with a mode that never exceeded 250 μm, except in March at station 2, when it is difficult to separate this subpopulation from the larger specimens. The main difference between the two stations was the higher proportion of larger individuals (>400 μm) at station 2, which

was visible through the better-developed tails at the right side of the distribution graphs. The low number of *Ammonia* sp. T6 individuals at station 1 did not allow us to draw any firm conclusion concerning the size distribution at this station (Supplementary Figure 3). At station 2, a group of individuals with smaller diameters (< 300 μm) was always present. The overall size distribution showed a clear shift to higher diameters between March (median = 279 μm) and May (median = 373 μm, Fig. 7), which is also evidenced by the much higher proportion of larger individuals. Specimens larger than 400 μm were abundantly found until November (median = 378 μm), but started to diminish in December, as is also shown by the decrease of the median to 339 μm. Our tentative to distinguish cohorts by using a deconvolution method to separate the total size distributions into a sum of Gaussian curves was not conclusive. The main problem was the fact that we did not have any information concerning individuals smaller than 125 μm, so that our size distributions were systematically skewed on the left side (i.e. toward small individuals). An additional problem was the large number of smaller specimens which were always present. Because the identification of individual cohorts was not successful, parameters like reproduction rate, growth rate or lifespan were not assessable. Nevertheless, the size distribution data give some clues concerning the population dynamics of the two dominant species.

---

## Author Comment (AC3) · 2 Dec 2019

Changes made in our manuscript and in the supplementary material after referee'scomments/suggestions.

Please also note the supplement to this comment:
https://www.biogeosciences-discuss.net/bg-2019-382/bg-2019-382-AC3-supplement.pdf

---

## Author Response (AR2)

Dear Dr Hiroshi Kitazato,

Thank you for giving me the opportunity to read the revised version of Richirt et al's MS resubmitted to Biogeosciences. I have read the revised version, the comments from the two reviewers and the response letters and I think the authors have done a good job revising the MS. However I have a few comments of more technical nature, which I hope the authors correct before publication.

The introduction and the aim are now well described, however, I lack a map of the study site. In particular, since this study is very much focused on one particular area.

As suggested by the reviewer, we added a map as figure 1 in the revised manuscript (with reference line 150).

On row 162, there should be one centimetre, to make clearer it was 0-1 cm that was studied.

We added between brackets that we considered the first centimetre (line 162):

*"For each replicate, the uppermost centimetre (0–1 cm) of the core was then transferred on board in a vial of 250 mL, and 30 mL of seawater (at the same temperature than in situ) was added in the vial."*

row 191-195 can be excluded, it doesn't move the text forward that the authors highlight that molecular data for *E. magellancium* is lacking. One only wonders why they didn't do the analyses then.

We would prefer to keep the statement about our attribution of *E. magellanicum* formal name to our specimens. In our opinion, this emphasizes two important things:

1- Contrary to *Ammonia* sp. T6 and *Elphidium selseyense*, we have no molecular data about assignation of these specimens to this formal name, thus our taxonomic attribution is less certain.

2- No fresh live material was collected at the time of sampling, therefore it was not possible to sequence *E. magellanicum* from Grevelingen. In addition, molecular analyses on what we usually name *E. magellanicum*, publicly unavailable yet (no

*sequences published in our knowledge), are needed to better constraint what E. magellanicum in reality is.*

rows 197-205 could be shortened, the Elphidium discussion is already in Darling et al., 2016 and no need to be repeated here.

As proposed by the reviewer we shortened this paragraph (lines 197-201).

The paragraph regarding size distribution should be shortened, in particular since the data is only reported as SI. Also figure 1 should be moved to SI1. There is no point to have the first 1 showing methods that are now moved to SI. I fully recognize the work that has been put into all those measurements.

As proposed by the reviewer we shortened as much as possible this section (section 2.5 in the manuscript, lines 207-230) without omitting information. We also moved figure 1 of the original manuscript to the supplementary material, which then became Supplementary Figure 2.

In the result section, I recommend to decrease the significant figures and avoid decimal numbers. What does 1.1 specimens/10 cc or 449.9 mean? it makes more sense to present 1 and 450 specimens /10cc. The same holds for per cent. The should be a space between the number and %.

It is evident that values such as 0.1 individual/10 cc do not make sense from a biological point of view. However, these data are statistical averages, standardised for 10 cc, which are systematically accompanied by a standard deviation, based on the comparison of two replicate samples. It is customary to give such statistical values with one decimal. The same is true for the percentage values. Next, in case of comparison of standardised data based on very different sediment volumes, the loss of the decimal value could lead to large errors. For this reason we would like to keep our numbers like they are.

We deleted % in table 2 and specified in the caption that the numbers in brackets are relative abundances in %.

Is Table 1 necessary?, the sampling date can be included in table 2.

The reviewer is right that table 1 could be merged with table 2. We did not merged them in the original manuscript for two reasons:

1- This would give redundant information if added to table 2 (because the two stations were sampled the same day, so two times the same day in the merged table) and this would mean to add a column where we have only double crosses (because 2 cores were sampled at all moment of sampling).

2- Table 1 in its current form allows the reader to quickly get the day of sampling for each month and compare the two stations very easily regarding the number of cores sampled.

For these two reasons we would like to keep these two tables separated in our manuscript.

In Table 2, including the unit in the header row and only present numbers (exclude % unit). In table 2 it is unnecessary to include 100% in the final column, I would also no call it total assemblage as it is total concentration.

We deleted the % in table 2 and specified in the caption the numbers in brackets are relative abundances in %.

We deleted the 100% in the last column and changed the header from "Total assemblage" to "Total". We changed the caption accordingly.

We also corrected some mistakes in the values of this table after careful double-checking.
In the SI table 3, the total concentration needs to be included or the total number of counted foraminifera for each sample.

We added a column "Total" as suggested by the reviewer in the Supplementary Table 3. We changed the caption accordingly.

I'm looking forward to seeing the MS in print as it includes some very interesting data from low O2 environments.

[revised manuscript text omitted]

**STATION 1**

| Species | | Elphidium selseyense | | Ammonia sp. T6 | | Elphidium magellanicum | | Trochammina inflata | | Total assemblage | |
|---|---|---|---|---|---|---|---|---|---|---|---|
| Year | Month | A | B | A | B | A | B | A | B | A | B |
| 2011 | August | 2.1 | 0.4 | 1.4 | 1.1 | 0.0 | 0.0 | 0.0 | 0.0 | 4.2 | 2.5 |
| 2011 | November | 0.0 | 1.1 | 0.0 | 0.7 | 0.0 | 0.0 | 0.0 | 0.0 | 0.0 | 2.1 |
| 2012 | January | 2.8 | 7.4 | 0.7 | 5.7 | 0.0 | 0.4 | 0.4 | 2.1 | 5.0 | 18.0 |
| 2012 | March | 28.6 | 19.1 | 12.0 | 13.8 | 29.4 | 13.8 | 2.1 | 0.7 | 75.7 | 48.5 |
| 2012 | May | 141.5 | 531.6 | 13.8 | 4.6 | 63.0 | 129.8 | 0.4 | 3.2 | 222.1 | 677.6 |
| 2012 | July | 76.0 | 247.9 | 8.1 | 12.4 | 3.9 | 3.5 | 0.0 | 0.0 | 88.4 | 270.6 |
| 2012 | September | 21.2 | 38.2 | 0.7 | 3.9 | 0.0 | 0.0 | 0.0 | 0.7 | 21.9 | 46.0 |
| 2012 | November | 0.7 | 1.4 | 0.4 | 0.4 | 0.0 | 0.0 | 0.0 | 0.0 | 1.4 | 1.8 |

**STATION 2**

| Species | | Elphidium selseyense | | Ammonia sp. T6 | | Elphidium magellanicum | | Trochammina inflata | | Total assemblage | |
|---|---|---|---|---|---|---|---|---|---|---|---|
| Year | Month | A | B | A | B | A | B | A | B | A | B |
| 2011 | August | 53.8 | 95.8 | 72.5 | 91.6 | 0.0 | 0.0 | 10.6 | 18.7 | 140.1 | 208.0 |
| 2011 | November | 33.2 | 71.4 | 61.9 | 59.8 | 0.0 | 0.0 | 13.1 | 10.6 | 111.1 | 146.4 |
| 2012 | January | 122.0 | 201.6 | 263.1 | 189.2 | 1.1 | 0.7 | 142.5 | 100.4 | 545.4 | 501.9 |
| 2012 | March | 225.6 | 203.7 | 275.2 | 152.8 | 41.0 | 56.6 | 73.9 | 76.0 | 624.2 | 500.5 |
| 2012 | May | 254.6 | 321.8 | 165.9 | 128.4 | 120.6 | 111.4 | 42.1 | 30.1 | 602.3 | 607.3 |
| 2012 | July | 318.3 | 246.9 | 172.2 | 144.7 | 39.6 | 36.1 | 35.4 | 27.6 | 589.9 | 473.2 |
| 2012 | September | 415.2 | 315.8 | 141.1 | 63.7 | 97.3 | 46.7 | 14.9 | 17.3 | 681.2 | 453.8 |
| 2012 | October | 104.7 | 92.7 | 87.0 | 111.1 | 2.1 | 1.4 | 5.3 | 9.5 | 205.8 | 217.2 |
| 2012 | November | 29.4 | 32.5 | 66.5 | 29.7 | 3.9 | 4.2 | 5.0 | 2.5 | 108.9 | 73.2 |
| 2012 | December | 281.2 | 223.2 | 78.9 | 77.1 | 16.3 | 34.7 | 15.9 | 9.5 | 405.3 | 350.5 |

**Supplementary Table 3.** Living foraminiferal abundances for each replicate, year and month for all the species of the assemblage (ind.  **10cm⁻³). Empty cases represent the absence in the sample. Last column: absolute abundance of the total fauna.**

| Year | Station | Replicate | Month | *Ammonia falsobeccarii (T15)* | *Ammonia sp. T1* | *Ammonia sp. T2* | *Ammonia sp. T3* | *Ammonia sp. T6* | *Bulimina denudata* | *Bulimina elongata* | *Bulimina marginata* | *Bulimina sp.* | *Cassidulina sp.* | *Elphidium selseyense* | *Elphidium magellanicum* | *Elphidium magellanicum (encrusted)* | *Elphidium margaritaceum* | *Elphidium sp.* | *Epistominella sp.* | *Haynesina depressula* | *Haynesina germanica* | *Hopkinsina sp.* | *Leptohalysis sp.* | *Non determined* | *Nonion sp.* | *Nonionella sp.* | *Quinqueloculina leavigata* | *Quinqueloculina sp.* | *Stainforthia sp.* | *Textularia sp.* | *Trochammina inflata* | Total |
|---|---|---|---|---|---|---|---|---|---|---|---|---|---|---|---|---|---|---|---|---|---|---|---|---|---|---|---|---|---|---|---|---|
| 2011 | 1 | A | August | 0.0 | 0.0 | 0.0 | 0.0 | 1.4 | 0.0 | 0.0 | 0.0 | 0.0 | 0.0 | 2.1 | 0.0 | 0.0 | 0.0 | 0.0 | 0.0 | 0.0 | 0.0 | 0.0 | 0.0 | 0.0 | 0.4 | 0.0 | 0.0 | 0.4 | 0.0 | 0.0 | 0.0 | 4.2 |
| 2011 | 1 | A | November | 0.0 | 0.0 | 0.0 | 0.0 | 0.0 | 0.0 | 0.0 | 0.0 | 0.0 | 0.0 | 0.0 | 0.0 | 0.0 | 0.0 | 0.0 | 0.0 | 0.0 | 0.0 | 0.0 | 0.0 | 0.0 | 0.0 | 0.0 | 0.0 | 0.0 | 0.0 | 0.0 | 0.0 |  |
| 2012 | 1 | A | January | 0.0 | 0.0 | 0.0 | 0.0 | 0.7 | 0.0 | 0.0 | 0.0 | 0.0 | 0.0 | 2.8 | 0.0 | 0.0 | 0.0 | 0.0 | 0.0 | 1.1 | 0.0 | 0.0 | 0.0 | 0.0 | 0.0 | 0.0 | 0.0 | 0.0 | 0.0 | 0.0 | 0.4 | 5.0 |
| 2012 | 1 | A | March | 0.4 | 0.0 | 1.1 | 0.0 | 12.0 | 0.4 | 0.0 | 0.0 | 0.0 | 0.0 | 28.6 | 29.4 | 0.0 | 0.0 | 0.4 | 0.0 | 0.4 | 0.0 | 0.0 | 0.0 | 0.0 | 0.0 | 0.0 | 0.0 | 0.7 | 0.0 | 0.4 | 2.1 | 75.7 |
| 2012 | 1 | A | May | 0.0 | 0.0 | 0.0 | 0.0 | 13.8 | 1.1 | 0.0 | 0.4 | 0.0 | 0.0 | 141.5 | 47.7 | 15.2 | 0.0 | 0.0 | 0.0 | 0.0 | 0.0 | 0.0 | 0.0 | 0.0 | 0.4 | 0.0 | 0.4 | 1.1 | 0.0 | 0.4 | 0.4 | 222.1 |
| 2012 | 1 | A | July | 0.0 | 0.0 | 0.0 | 0.0 | 8.1 | 0.0 | 0.0 | 0.0 | 0.0 | 0.0 | 76.0 | 1.8 | 2.1 | 0.0 | 0.0 | 0.0 | 0.0 | 0.0 | 0.0 | 0.0 | 0.0 | 0.0 | 0.0 | 0.0 | 0.0 | 0.0 | 0.4 | 0.0 | 88.4 |
| 2012 | 1 | A | September | 0.0 | 0.0 | 0.0 | 0.0 | 0.7 | 0.0 | 0.0 | 0.0 | 0.0 | 0.0 | 21.2 | 0.0 | 0.0 | 0.0 | 0.0 | 0.0 | 0.0 | 0.0 | 0.0 | 0.0 | 0.0 | 0.0 | 0.0 | 0.0 | 0.0 | 0.0 | 0.0 | 0.0 | 21.9 |
| 2012 | 1 | A | November | 0.0 | 0.0 | 0.4 | 0.0 | 0.4 | 0.0 | 0.0 | 0.0 | 0.0 | 0.0 | 0.7 | 0.0 | 0.0 | 0.0 | 0.0 | 0.0 | 0.0 | 0.0 | 0.0 | 0.0 | 0.0 | 0.0 | 0.0 | 0.0 | 0.0 | 0.0 | 0.0 | 0.0 | 1.4 |
| 2011 | 1 | B | August | 0.0 | 0.0 | 0.0 | 0.0 | 1.1 | 0.0 | 0.0 | 0.0 | 0.0 | 0.0 | 0.4 | 0.0 | 0.0 | 1.1 | 0.0 | 0.0 | 0.0 | 0.0 | 0.0 | 0.0 | 0.0 | 0.0 | 0.0 | 0.0 | 0.0 | 0.0 | 0.0 | 0.0 | 2.5 |
| 2011 | 1 | B | November | 0.0 | 0.0 | 0.0 | 0.0 | 0.7 | 0.0 | 0.0 | 0.0 | 0.0 | 0.0 | 1.1 | 0.0 | 0.0 | 0.0 | 0.0 | 0.0 | 0.0 | 0.0 | 0.0 | 0.0 | 0.0 | 0.0 | 0.0 | 0.0 | 0.4 | 0.0 | 0.0 | 0.0 | 2.1 |
| 2012 | 1 | B | January | 0.0 | 0.0 | 0.7 | 0.0 | 5.7 | 0.0 | 0.0 | 0.0 | 0.0 | 0.0 | 7.4 | 0.4 | 0.0 | 0.4 | 0.0 | 0.0 | 0.0 | 0.0 | 0.0 | 0.0 | 0.0 | 0.0 | 0.0 | 1.1 | 0.4 | 0.0 | 0.0 | 2.1 | 18.0 |
| 2012 | 1 | B | March | 0.0 | 0.0 | 0.0 | 0.0 | 13.8 | 0.0 | 0.0 | 0.0 | 0.0 | 0.0 | 19.1 | 13.8 | 0.0 | 0.0 | 0.0 | 0.0 | 0.4 | 0.0 | 0.0 | 0.0 | 0.4 | 0.0 | 0.0 | 0.4 | 0.0 | 0.0 | 0.0 | 0.7 | 48.5 |
| 2012 | 1 | B | May | 0.0 | 0.0 | 0.0 | 0.0 | 4.6 | 0.4 | 0.0 | 0.0 | 0.0 | 0.0 | 531.6 | 93.4 | 36.4 | 0.4 | 0.0 | 0.7 | 0.4 | 0.0 | 0.0 | 0.0 | 2.1 | 0.0 | 0.4 | 0.4 | 1.1 | 2.5 | 0.4 | 3.2 | 677.6 |
| 2012 | 1 | B | July | 0.0 | 0.0 | 0.4 | 0.0 | 12.4 | 0.4 | 0.0 | 0.7 | 0.0 | 0.0 | 247.9 | 2.1 | 1.4 | 1.4 | 0.0 | 0.4 | 0.0 | 0.0 | 0.0 | 0.0 | 0.0 | 0.0 | 0.7 | 0.7 | 0.4 | 1.8 | 0.0 | 0.0 | 270.6 |
| 2012 | 1 | B | September | 0.0 | 0.0 | 0.0 | 0.0 | 3.9 | 0.0 | 0.0 | 0.0 | 0.0 | 0.0 | 38.2 | 0.0 | 0.0 | 0.4 | 0.0 | 0.0 | 0.0 | 0.0 | 0.0 | 0.0 | 0.0 | 0.0 | 0.0 | 0.0 | 0.0 | 2.5 | 0.4 | 0.7 | 46.0 |
| 2012 | 1 | B | November | 0.0 | 0.0 | 0.0 | 0.0 | 0.4 | 0.0 | 0.0 | 0.0 | 0.0 | 0.0 | 1.4 | 0.0 | 0.0 | 0.0 | 0.0 | 0.0 | 0.0 | 0.0 | 0.0 | 0.0 | 0.0 | 0.0 | 0.0 | 0.0 | 0.0 | 0.0 | 0.0 | 0.0 | 1.8 |
| 2011 | 2 | A | August | 0.0 | 0.0 | 0.0 | 0.0 | 72.5 | 0.0 | 0.0 | 0.0 | 0.0 | 0.0 | 53.8 | 0.0 | 0.0 | 0.7 | 0.0 | 0.0 | 0.0 | 0.0 | 0.0 | 0.0 | 0.4 | 0.0 | 1.1 | 0.4 | 0.4 | 0.0 | 0.4 | 10.6 | 140.1 |
| 2011 | 2 | A | November | 0.0 | 0.0 | 0.0 | 0.0 | 61.9 | 0.0 | 0.0 | 0.0 | 0.0 | 0.0 | 33.2 | 0.0 | 0.0 | 0.7 | 0.0 | 0.0 | 0.0 | 0.0 | 0.0 | 0.0 | 0.0 | 1.1 | 0.0 | 1.1 | 0.0 | 0.0 | 0.0 | 13.1 | 111.1 |
| 2012 | 2 | A | January | 0.7 | 0.0 | 2.5 | 8.8 | 263.1 | 0.0 | 1.1 | 0.0 | 0.0 | 0.0 | 122.0 | 1.1 | 0.0 | 0.7 | 0.4 | 0.0 | 1.1 | 0.0 | 0.0 | 0.0 | 0.0 | 0.7 | 0.4 | 0.0 | 0.0 | 0.0 | 0.4 | 142.5 | 545.4 |
| 2012 | 2 | A | March | 0.0 | 0.0 | 1.4 | 0.0 | 275.2 | 0.0 | 0.0 | 0.0 | 1.8 | 0.0 | 225.6 | 40.0 | 1.1 | 0.4 | 0.0 | 0.0 | 0.4 | 0.0 | 0.0 | 0.0 | 0.0 | 0.7 | 0.7 | 0.0 | 1.4 | 0.0 | 1.8 | 73.9 | 624.2 |
| 2012 | 2 | A | May | 0.0 | 0.0 | 1.1 | 0.0 | 165.9 | 0.0 | 0.0 | 0.4 | 0.0 | 3.9 | 254.6 | 38.6 | 82.1 | 0.4 | 0.0 | 0.0 | 1.4 | 0.0 | 0.0 | 0.0 | 0.0 | 0.0 | 3.2 | 0.4 | 2.1 | 1.4 | 5.0 | 42.1 | 602.3 |
| 2012 | 2 | A | July | 0.0 | 0.0 | 1.8 | 0.0 | 172.2 | 6.0 | 2.1 | 0.4 | 0.4 | 0.0 | 318.3 | 3.9 | 35.7 | 1.4 | 0.0 | 0.4 | 0.7 | 0.0 | 0.0 | 0.0 | 0.0 | 0.0 | 0.4 | 0.0 | 7.1 | 1.8 | 2.1 | 35.4 | 589.9 |

Cellules insérées

| Year | Q | Site | Month | c1 | c2 | c3 | c4 | c5 | c6 | c7 | c8 | c9 | c10 | c11 | c12 | c13 | c14 | c15 | c16 | c17 | c18 | c19 | c20 | c21 | c22 | c23 | c24 | c25 | c26 | c27 | c28 | Total |
|---|---|---|---|---|---|---|---|---|---|---|---|---|---|---|---|---|---|---|---|---|---|---|---|---|---|---|---|---|---|---|---|---|
| 2012 | 2 | A | September | 0.0 | 0.7 | 0.0 | 0.0 | 141.1 | 0.0 | 1.4 | 0.4 | 0.0 | 0.0 | 415.2 | 16.3 | 81.0 | 0.4 | 0.4 | 3.2 | 0.0 | 0.0 | 1.4 | 0.0 | 0.0 | 0.0 | 0.0 | 0.0 | 0.4 | 1.4 | 3.2 | 14.9 | 681.2 |
| 2012 | 2 | A | October | 0.0 | 0.4 | 0.7 | 0.0 | 87.0 | 1.1 | 2.5 | 0.4 | 0.0 | 0.0 | 104.7 | 0.0 | 2.1 | 0.0 | 0.0 | 0.0 | 0.0 | 0.0 | 0.0 | 0.4 | 0.0 | 0.0 | 0.0 | 0.0 | 0.0 | 0.0 | 1.4 | 5.3 | 205.8 |
| 2012 | 2 | A | November | 0.0 | 0.0 | 0.0 | 0.0 | 66.5 | 0.7 | 0.0 | 0.4 | 0.0 | 0.0 | 29.4 | 0.0 | 3.9 | 0.4 | 0.0 | 0.0 | 0.0 | 0.0 | 2.1 | 0.0 | 0.0 | 0.0 | 0.0 | 0.0 | 0.0 | 0.7 | 0.0 | 5.0 | 108.9 |
| 2012 | 2 | A | December | 0.7 | 0.0 | 1.8 | 0.0 | 78.9 | 1.1 | 0.7 | 1.4 | 0.0 | 0.0 | 281.2 | 0.4 | 15.9 | 0.0 | 0.0 | 0.7 | 0.4 | 0.0 | 1.8 | 0.0 | 0.4 | 0.0 | 0.4 | 0.0 | 0.4 | 0.4 | 3.2 | 15.9 | 405.3 |
| 2011 | 2 | B | August | 0.0 | 0.0 | 0.0 | 0.0 | 91.6 | 0.0 | 0.0 | 0.0 | 0.4 | 0.0 | 95.8 | 0.0 | 0.0 | 0.0 | 0.0 | 0.7 | 0.4 | 0.0 | 0.0 | 0.4 | 0.0 | 0.4 | 0.0 | 0.0 | 0.0 | 0.0 | 0.0 | 18.7 | 208.0 |
| 2011 | 2 | B | November | 0.0 | 0.0 | 0.0 | 0.0 | 59.8 | 0.0 | 0.0 | 0.0 | 0.4 | 0.0 | 71.4 | 0.0 | 0.0 | 1.1 | 0.0 | 0.0 | 1.1 | 0.0 | 0.0 | 0.0 | 1.1 | 0.0 | 0.0 | 0.0 | 1.1 | 0.0 | 0.0 | 10.6 | 146.4 |
| 2012 | 2 | B | January | 0.0 | 0.4 | 2.1 | 0.0 | 189.2 | 0.0 | 0.4 | 0.0 | 0.0 | 0.0 | 201.6 | 0.7 | 0.0 | 0.0 | 1.1 | 0.0 | 0.0 | 0.0 | 0.0 | 0.0 | 0.0 | 0.0 | 0.0 | 0.0 | 5.7 | 0.0 | 0.4 | 100.4 | 501.9 |
| 2012 | 2 | B | March | 0.0 | 0.0 | 1.1 | 0.0 | 152.8 | 0.4 | 0.0 | 0.0 | 2.1 | 0.0 | 203.7 | 56.2 | 0.4 | 1.1 | 0.7 | 1.4 | 0.0 | 0.0 | 0.0 | 0.0 | 0.0 | 1.1 | 0.4 | 0.0 | 1.8 | 0.7 | 0.7 | 76.0 | 500.5 |
| 2012 | 2 | B | May | 0.0 | 0.0 | 1.4 | 0.0 | 128.4 | 2.1 | 0.0 | 0.7 | 0.0 | 0.4 | 321.8 | 25.8 | 85.6 | 0.0 | 0.0 | 0.4 | 0.4 | 0.0 | 0.0 | 0.0 | 1.8 | 0.0 | 2.8 | 1.1 | 0.7 | 1.1 | 2.8 | 30.1 | 607.3 |
| 2012 | 2 | B | July | 0.0 | 1.1 | 1.4 | 0.0 | 144.7 | 0.4 | 1.8 | 1.8 | 2.1 | 0.0 | 246.9 | 8.1 | 27.9 | 0.7 | 0.0 | 1.1 | 1.1 | 0.0 | 0.0 | 0.0 | 0.0 | 0.0 | 0.4 | 2.1 | 1.1 | 0.7 | 2.5 | 27.6 | 473.2 |
| 2012 | 2 | B | September | 0.0 | 0.0 | 0.4 | 0.0 | 63.7 | 1.8 | 0.7 | 0.0 | 0.0 | 0.0 | 315.8 | 8.1 | 38.6 | 1.4 | 0.4 | 2.1 | 0.0 | 0.4 | 0.0 | 0.0 | 0.0 | 0.0 | 0.0 | 0.0 | 0.4 | 1.4 | 1.4 | 17.3 | 453.8 |
| 2012 | 2 | B | October | 0.0 | 0.7 | 1.1 | 0.0 | 111.1 | 0.4 | 0.0 | 0.0 | 0.0 | 0.0 | 92.7 | 1.1 | 0.4 | 0.0 | 0.0 | 0.4 | 0.0 | 0.0 | 0.0 | 0.0 | 0.0 | 0.0 | 0.0 | 0.0 | 0.0 | 0.0 | 0.0 | 9.5 | 217.2 |
| 2012 | 2 | B | November | 0.0 | 0.0 | 0.4 | 0.0 | 29.7 | 1.1 | 0.0 | 0.4 | 0.0 | 0.0 | 32.5 | 1.8 | 2.5 | 0.4 | 0.0 | 0.7 | 0.0 | 0.0 | 0.0 | 0.0 | 0.0 | 0.0 | 0.0 | 0.4 | 0.4 | 0.7 | | 2.5 | 73.2 |
| 2012 | 2 | B | December | 0.0 | 0.0 | 0.0 | 0.0 | 77.1 | 1.4 | 0.7 | 0.0 | 0.0 | 0.0 | 223.2 | 5.7 | 29.0 | 1.1 | 0.0 | 1.4 | 0.0 | 0.0 | 0.4 | 0.0 | 0.0 | 0.0 | 0.0 | 0.4 | 0.4 | 0.0 | 0.4 | 9.5 | 350.5 |

[Figure]

**Supplementary Figure 1. SEM images of spiral side and a 1000x magnification of the penultimate chamber for four individuals from Grevelingen station 1 identified T6 by molecular identification.**

[Figure]

**Supplementary Figure 2**

[Figure]

| Raw picture | Foraminifera isolation | Individual measurement |

**Supplementary Figure 2: Numerical treatment used for the size measurement for each image performed with ImageJ software. The three size fractions (125–150, 150–315, >315 µm) were analysed together for the size distribution analyses. The left figure shows the untreated image, the middle figure presents the next step, when all individual foraminifera are depicted. Finally, the figure on the right shows the individual foraminiferal outlines which were measured.**

[Figure]

**Supplementary Figure 3**: A: size distribution (maximum diameter for each individual in µm) of *Elphidium selseyense* for stations 1 (left) and 2 (right) in 2012. B: size distribution (maximum diameter for each individual in µm) of *Ammonia* sp. T6 for stations 1 (left) and 2 (right) in 2012. For each month, the number of individuals (n), the mode and the number of individuals associated to the mode (between brackets) are indicated in black. The medians are indicated by the red bars in each panel. In order to base our analysis on a sufficiently high number of specimens, we focused on *E. selseyense* and *Ammonia* sp. T6. As explained before, we only considered specimens retained on a 125 µm mesh meaning that juvenile specimens are not represented. Only the samples taken in 2012 were considered. The size distribution of *E. selseyense* was relatively similar between the two stations regarding the median, ranging from 253 µm (in May) to 295 µm (in November) at station 1 and from 261 µm (in October) to 290 µm (in March) at station 2. At both stations, we observed the presence of an abundant group of smaller specimens, with a mode that never exceeded 250 µm, except in March at station 2, when it is difficult to separate this subpopulation from the larger specimens. The main difference between the two stations was the higher proportion of larger individuals (>400 µm) at station 2, which

was visible through the better-developed tails at the right side of the distribution graphs. The low number of *Ammonia* sp. T6 individuals at station 1 did not allow us to draw any firm conclusion concerning the size distribution at this station (Supplementary Figure 3). At station 2, a group of individuals with smaller diameters (< 300 μm) was always present. The overall size distribution showed a clear shift to higher diameters between March (median = 279 μm) and May (median = 373 μm, Fig. 7), which is also evidenced by the much higher proportion of larger individuals. Specimens larger than 400 μm were abundantly found until November (median = 378 μm), but started to diminish in December, as is also shown by the decrease of the median to 339 μm. Our tentative to distinguish cohorts by using a deconvolution method to separate the total size distributions into a sum of Gaussian curves was not conclusive. The main problem was the fact that we did not have any information concerning individuals smaller than 125 μm, so that our size distributions were systematically skewed on the left side (i.e. toward small individuals). An additional problem was the large number of smaller specimens which were always present. Because the identification of individual cohorts was not successful, parameters like reproduction rate, growth rate or lifespan were not assessable. Nevertheless, the size distribution data give some clues concerning the population dynamics of the two dominant species.